



# HOMs and SOA formation from the oxidation of α- and β-phellandrenes by NO₃ radicals

Sergio Harb[1,2], Manuela Cirtog[1], Stéphanie Alage[1], Christopher Cantrell[1,3], Mathieu Cazaunau[1], Vincent Michoud[4], Edouard Pangui[1], Antonin Bergé[1], Chiara Giorio[5], Francesco Battaglia[3], and Bénédicte Picquet-Varrault[1]

[1] Univ Paris Est Creteil and Université de Paris Cité, CNRS, LISA, F-94010 Créteil, France
[2] Institut National de l'Environnement Industriel et des Risques (INERIS), Verneuil-en-Halatte, 60550, France
[3] University of Colorado, Atmospheric and Oceanic Sciences (ATOC), Boulder, CO 80309
[4] Université Paris cité and Univ Paris Est Creteil, CNRS, LISA, F-75013 Paris, France
[5] Yusuf Hamied Department of Chemistry, University of Cambridge, Lensfield Road, Cambridge, CB2 1EW, United Kingdom

*Correspondence to*: sergio.harb@ineris.fr; benedicte.picquet-varrault@lisa.ipsl.fr

**Abstract.** Nighttime $NO_3$-initiated oxidation of monoterpenes plays a crucial role as source of organic nitrates (ONs) and secondary organic aerosols (SOA), impacting climate, air quality, and human health. Nevertheless, monoterpene reactions with $NO_3$ remain poorly understood. This study provides an in-depth investigation of the $NO_3$-initiated oxidation of α- and β-phellandrenes, through simulation chamber experiments and a combination of various analytical techniques (FTIR, PTR-ToF-MS, ACSM, nitrate-CI-APi-ToF-MS, Orbitrap, SMPS). SOA yields were measured, and oxidation products, including highly oxygenated organic molecules (HOMs), were investigated in gas and aerosol phases. Numerical simulations were also performed to investigate the dominant chemical regimes for $RO_2$ radicals. We found that α- and β-phellandrenes are efficient SOA precursors with yields reaching up to 35 % and 60 %, respectively, with β-phellandrene generating significantly more SOA than α-phellandrene. Both monoterpenes produce large amounts of ONs in gas and aerosol phases with total molar yields of 40-60 %. Similar gas-phase products were detected for α- and β-phellandrenes. In particular, carbonyl nitrates, dicarbonyl nitrates and dicarbonyls were detected as first-generation products. Autooxidation processes were also shown to occur with numerous gas-phase HOM monomers and dimers detected. Chemical mechanisms have been proposed to explain products formation. Since gas-phase products were similar for both monoterpenes, they do not explain the differences in SOA yields. However, some differences in aerosol-phase composition were observed which may explain why β-phellandrene is a more efficient SOA precursor. This study is the first mechanistic investigation of the reactions of α- and β-phellandrenes with $NO_3$ radical.

## 1. Introduction

Biogenic volatile organic compounds (BVOCs) are emitted in significant amounts into the atmosphere by forests and crops, and represent 75 % to 90 % of global non-methane VOC emissions (Goldstein and Galbally, 2007; Guenther et al., 2006, 2012; Messina et al., 2016; Sindelarova et al., 2014). Monoterpenes (MTs), which are unsaturated $C_{10}H_{16}$ compounds, are the second




most abundant family of species among BVOCs, after isoprene. Under ambient atmospheric conditions, MTs are primarily oxidized by major atmospheric oxidants such as hydroxyl radical (OH), ozone ($O_3$), and nitrate radical ($NO_3$) resulting in short lifetimes of only few minutes for the most reactive ones.

The $NO_3$ radical is mainly formed in the atmosphere through the reaction of nitrogen dioxide ($NO_2$) with $O_3$. This radical is mostly abundant during the night due to its rapid photolysis (Brown and Stutz, 2012). It is also highly reactive towards nitric oxide (NO) and unsaturated BVOCs, such as isoprene and monoterpenes, making it a major nighttime oxidant for these compounds (Liebmann et al., 2019; Mogensen et al., 2015), particularly in mixed atmospheres, i.e. influenced by both biogenic and anthropogenic emissions (Brown et al., 2011, 2009; Geyer et al., 2001).

$NO_3$-initiated oxidation mechanisms of MTs have been less investigated than reactions with OH and $O_3$ (Ng et al., 2017). Recently, this chemistry has received increased attention in the literature due to its potential influences on climate, air quality, and health through the formation of many gaseous and particulate secondary pollutants. One important group of products resulting from these reactions is organic nitrates (ONs). These compounds play a significant role as temporary reservoirs for reactive nitrogen by undergoing long-range transport in the free troposphere before decomposing and releasing $NO_x$ (NO +

$NO_2$) in remote regions. This process influences the Nitrogen cycle and affects the regional $O_3$ budget (Fisher et al., 2016; Ito et al., 2007). Moreover, multifunctional ONs have low volatility, allowing them to partition into the condensed phase, and thereby to contribute to the formation and growth of secondary organic aerosols (SOA). Recent field campaigns have provided insights into the aerosol chemical composition, revealing substantial contributions of ONs to the total mass of organic aerosols (OA), ranging from 10 % to 75 % (e.g. Huang et al., 2019; Kenagy et al., 2021; Kiendler-Scharr et al., 2016; Lee et al., 2016;

Xu et al., 2015).

Among the numerous monoterpenes emitted into the atmosphere, only a few of them have been subject to experimental studies to investigate the mechanisms and the SOA formation from oxidation by $NO_3$. Most of these studies were performed in simulation chambers and on the most common monoterpenes whose chemistry has already been well-studied, namely α-pinene, β-pinene, $\Delta^3$-carene, and limonene (e.g. Draper et al., 2019, 2021; Shen et al., 2021; Bates et al., 2022; Day et al.,

2022; DeVault et al., 2022; Mutzel et al., 2021). Only a few studies were performed on other monoterpenes for which no data was previously available, e.g. α- and γ-terpinene (Fouqueau et al., 2020; Slade et al., 2017) and α-thujene (Dam et al., 2022). In most cases, laboratory studies have demonstrated that monoterpenes are efficient precursors of SOA with yields typically above 20 %, except for some of them, such as α-pinene. Indeed, studies have shown very low or no SOA formation from the oxidation of α-pinene by $NO_3$, with yields ranging from 0 to 13 % (Fry et al., 2014; Hallquist et al., 1999; Mutzel et al., 2021;

Nah Theodora et al., 2016; Spittler et al., 2006). Moreover, chamber studies have revealed that the SOA yields ($Y_{SOA}$) differ significantly among monoterpenes having minor structural differences. For example, α- and γ-terpinene, which differ only in the position of the double bond exhibit large differences in SOA yields under similar experimental conditions, ranging from 1.2 % (for α-terpinene) to 10 % (for γ-terpinene) for an aerosol mass loading of 10 µg m$^{-3}$ (Fouqueau et al., 2020a). Furthermore, different SOA yields were also observed for the same compound across different studies, depending on the





experimental conditions and the dominant chemical regime for peroxy ($RO_2$) radicals. As discussed in the review from Ng et al. (2017), the fate of peroxy radicals, in particular the competition between $RO_2$ + $RO_2$, $RO_2$ + $NO_3$ and $RO_2$ + $HO_2$, may affect the SOA yields as it determines the subsequent chemistry and leads to different types of products as shown in the following reactions (R1-5):

$$RO_2 + R^{'}O_2 \rightarrow ROH + R^{'}=O \tag{R1}$$

$$RO_2 + HO_2 \rightarrow ROOH \tag{R2}$$

$$RO_2 + R^{'}O_2 \rightarrow RO + R^{'}O + O_2 \tag{R3}$$

$$RO_2 + NO_3 \rightarrow RO + NO_2 + O_2 \tag{R4}$$

$$RO_2 + R^{'}O_2 \rightarrow ROOR^{'} + O_2 \tag{R5}$$

For example Bates et al. (2022) conducted a series of experiments on α-pinene under different chemical conditions. They found

low SOA yield values (3-12 %) when the $RO_2$ + $NO_3$ pathway was dominant, in good agreement with studies by Moldanova and Ljungström (2000); Mutzel et al. (2021) and Spittler et al. (2006). On the contrary, they did not observe any SOA formation through the $RO_2$ + $HO_2$ pathway, which is consistent with the study of Fry et al. (2014), and they found SOA yields > 21 % when the $RO_2$ + $RO_2$ was the dominant pathway. The $RO_2$ fate appears therefore to be a key step in the SOA formation and needs to be further investigated.

Recently, a new class of gas-phase organic compounds named highly oxygenated organic molecules (HOMs, which are organic compounds with a minimum of five to six oxygen atoms) have been observed in ambient air and laboratory experiments, and found to play a critical and significant role in both nucleation and SOA growth from the $NO_3$ oxidation of terpenes (Bell et al., 2022; Boyd et al., 2015; Claflin and Ziemann, 2018; Dam et al., 2022; Guo et al., 2022; Nah Theodora et al., 2016; Shen et al., 2021; Takeuchi and Ng, 2019). HOMs are primarily produced through the autoxidation process of peroxy radicals by

intramolecular H-shifts, followed by the addition of $O_2$, and resulting in the formation of a new, more oxidized $RO_2$ radical (Crounse et al., 2013; Ehn et al., 2014; Mentel et al., 2015; Møller et al., 2019; Vereecken and Nozière, 2020). This H-shift can proceed several times leading to highly oxygenated $RO_2$, which can then evolve towards the formation of stable closed-shell products following either bimolecular reactions (R1 to R4) or unimolecular termination channels, such as intramolecular H-abstraction from a carbon with an −OOH group, followed by OH loss (Bianchi et al., 2019). HOM monomers bearing

various oxygenated functional groups (hydroperoxide, carbonyl, hydroxyl) have been reported in several studies (e.g. Dam et al., 2022; Guo et al., 2022; Jokinen et al., 2015; Shen et al., 2022, 2021). Moreover, HOM dimers can be formed by the combination of two $RO_2$ radicals through accretion reactions (reaction R5) (Berndt et al., 2018a). Another pathway for HOM formation involves H-shifts within RO radicals, followed by the addition of $O_2$. These various pathways, along with others, have been comprehensively reviewed by Bianchi et al. (2019). Despite their potential contribution to SOA, studies of HOM

formation from the $NO_3$ initiated oxidation of MTs are still limited to few compounds including α-pinene, β-pinene, $\Delta^3$-carene, limonene, and α-thujene (e.g. Dam et al., 2022; Draper et al., 2019; Faxon et al., 2018; Guo et al., 2022; Mayorga et al., 2022; Shen et al., 2021), compared to HOM formation through oxidation by OH and $O_3$.



Given the limited number of studies on the NO$_3$-initiated oxidation of MTs and the wide range of measured SOA yields, new experimental studies are required covering a broader range of compounds to improve our understanding of these mechanisms

and to identify key factors that influence the SOA formation. Among all emitted monoterpenes, α-phellandrene has been identified as a major component in extracts and emissions from numerous Eucalyptus species (Li et al., 1995; Maghsoodlou et al., 2015; Maleknia et al., 2009; Pavlova et al., 2015), which are the most widely planted hardwood forest trees on the global scale (>20 million ha) (Myburg et al., 2014). β-phellandrene is a major contributor to emissions from coniferous trees, such as Scots pine and Norway spruce  (Hao et al., 2009; Janson, 1992; Joutsensaari et al., 2015; Ylisirniö et al., 2020).

In this paper, we have therefore investigated the reactivity of NO$_3$ radical with α- and β-phellandrenes by performing experiments in the CESAM (Multiphase Atmospheric Experimental Simulation Chamber, http://www.cesam.cnrs.fr/) chamber. Oxidation products, including HOM monomers and dimers, in both gas and aerosol phases were identified. SOA and total organic nitrates yields were also determined. Formation pathways for the detected products were proposed based on their time profiles and available information in the literature. Finally, the results for α- and β-phellandrenes were compared with

other monoterpenes having similar chemical structures.

## 2. Material and methods

### 2.1 Chamber facility and analytical devices

Experiments were carried out in the CESAM chamber, which is a large evacuable reactor that consists of a stainless-steel cylindrical vessel with a volume of 4177 L. It has been specifically designed to study multiphase atmospheric processes and

has been described in details by Wang et al. (2011). The chamber is equipped with a fan that stirs the chamber mixture leading to a mixing time of approximately 1 min. The lifetime of aerosols within the CESAM chamber is very long (up to 4 days, depending on the particle size and composition), thanks to low levels of electrostatic charges on the chamber's wall. It is therefore well-suited to investigate aerosol formation and aging (Lamkaddam et al., 2017).

Several analytical instruments were used to measure the gas-phase composition (reactants and oxidation products) in the

chamber. Measurement of oxidation products, including organic nitrates, was performed using two PTR-ToF-MS which were operated in two ionization modes, H$_3$O$^+$ and NO$^+$, and using low acceleration energy in the instrument reactor to limit fragmentation (Duncianu et al., 2017). These two ionization modes are complementary and enable cross-checking for the identification of products. In the H$_3$O$^+$ ionization mode, ONs were generally detected as quasi-molecular ions [M+H]$^+$ since ionization proceeds by protonation following reaction (R6). In NO$^+$ ionization mode, ONs were detected mainly as molecular

ions [M]$^+$ or as adducts [M+NO]$^+$ since ionization proceeds by charge transfer and by adduct formation following reactions (R7) and (R8), respectively.

$R + H_3O^+ \rightarrow RH^+ + H_2O$                                                                                          (R6)

$NO^+ + R \rightarrow R^+ + NO$                                                                                                (R7)



$$NO^+ + R \rightarrow R \cdot NO^+ \tag{R8}$$

A nitrate chemical ionization inlet (CI), coupled to an Atmospheric Pressure Interface-Time-of-Flight Mass Spectrometer (nitrate-CI-APi-ToF-MS here after named ToF-CIMS; Aerodyne Research Inc., and Tofwerk AG; with a mass resolution 4000 M/δM) was used to detect highly oxygenated molecules within the chamber. Sampling was carried out at a flow rate of 1 L min$^{-1}$ and then diluted with 6 L min$^{-1}$ of dry air before being sampled by the instrument. Briefly, the sample first undergoes a soft ionization at ambient pressure, using nitrate as the reagent ion, to form clusters between the reactant molecule and the

reagent ion ($HOM + NO_3^- \rightarrow HOM \cdot NO_3^-$). The chemically ionized sample is then guided by ion optics through the APi into the ToF, where molecules are separated and detected according to their mass-to-charge ratios (m/z) (Junninen et al., 2010). Data were obtained from the co-addition of scans averaged over 1 min. The ToF-CIMS data allows identification (providing molecular formulas) of gas-phase oxidation products. Instrument setup used during this studies is the same as described by Alage et al. (2024).

CESAM chamber is also equipped with an *in situ* long path Fourier Transform InfraRed spectrometer, (FTIR, Tensor 37, Bruker). It allows the acquisition of spectra in the 700-4000 cm$^{-1}$ range, with a resolution of 0.5 cm$^{-1}$ and an optical path length of 120 m. To quantify species of interest, the following integrated band intensities (IBI), expressed in cm molecule$^{-1}$ on a logarithmic base e scale, were used:

- $IBI_{N_2O_5}$(1205-1275 cm$^{-1}$) = $(3.9 \pm 0.2) \times 10^{-17}$ (Fouqueau et al., 2020)
- $IBI_{NO_2}$(1530-1680 cm$^{-1}$) = $(5.6 \pm 0.2) \times 10^{-17}$ (Rothman et al., 2003)
- $IBI_{HNO_3}$(840-930 cm$^{-1}$) = $(2.1 \pm 0.2) \times 10^{-17}$ (Hjorth et al., 1987)
- $IBI_{\alpha\text{-phellandrene}}$(2787-3068 cm$^{-1}$) = $(2.4 \pm 0.2) \times 10^{-17}$ (from this study)
- $IBI_{\beta\text{-phellandrene}}$(2787-3068 cm$^{-1}$) = $(2.5 \pm 0.2) \times 10^{-17}$ (from this study)

Furthermore, FTIR was used to measure the concentration of total ONs, assuming that all these species absorb at 850 cm$^{-1}$ and

that their absorption cross sections are similar regardless of the chemical structure of the considered organic nitrate. For this study, the IBI used to quantify ONs is $IBI_{ONs}$ (820-900 cm$^{-1}$) = $(9.5 \pm 2.9) \times 10^{-18}$ cm molecule$^{-1}$.

A CAPS (Cavity Attenuated Phase Shift) NO$_2$ analyzer (Model T500U, from Teledyne API), was used to monitor NO$_2$ with a limit of detection of 0.04 ppb for an integration time of 30 s.

Concerning the characterization of the particulate phase, the particle number size distribution was measured using a scanning

mobility particle sizer (SMPS, TSI Inc., DMA model 3081, CPC model 3772, 2.0/0.2 L min$^{-1}$ sheath/aerosol flow rates, 135 s time resolution) which allows the measurement of the particle number concentration within the electrical mobility diameter range of 20 to 880 nm. Moreover, a Tapered Element Oscillating Microbalance (TEOM; Series 1400a, Rupprecht and Patashnick, 5 min time resolution) was used for some experiments allowing the direct measurement of the particle mass concentration. The use of both TEOM and SMPS during the same experiments enabled to calculate the density of particles

which was found to be 1.45 g cm$^{-3}$. This value is consistent with the density reported in the literature for SOA formed by BVOC + NO$_3$ reactions (Fry et al., 2014; Boyd et al., 2015; Draper et al., 2015) and it was used to convert the particle number



size distributions into mass distributions. SOA mass loading and chemical composition, including ammonium, nitrate, chloride, sulfate, and organic species, were measured in real time using an Aerosol Chemical Speciation Monitor (ACSM, Aerodyne Research, Inc.). Measurements of particles with this instrument were possible within the aerodynamic diameter size range of

40 to 1000 nm. For more detailed chemical characterization of the particulate phase, samples were collected on quartz filters (Pallflex, Tissuquartz, Ø 47 mm) at a flow rate of 6 L min$^{-1}$ or 8 L min$^{-1}$ and for a duration of two to three hours following the oxidation of the BVOC. Prior to the sampling, the quartz filters have been carbonized at 600 °C during six hours to eliminate any possible organic contamination. To avoid condensation of gas-phase products onto the filters, a charcoal denuder was installed upstream of the filter sampling. All the collected sample filters as well as blank filters were stored at a temperature

below −20 °C before analysis. Two different techniques were used for analysis. The measurement of total ONs in the particulate phase was performed following the protocol described in Rindelaub et al. (2015) where SOA were extracted from filters using 5 mL of CCl$_4$ for 15 min in an ultrasonic bath. Then, the sample was analyzed by FTIR (Perkin Elmer Spectrum 2). The quantification of total ONs in the liquid phase was performed using the IBI$_{ONs}$(1264-1310 cm$^{-1}$) = 5570 ± 1100 L mol$^{-1}$ cm$^{-2}$. The aerosol chemical composition at the molecular scale was determined by using a high-resolution (HR) LTQ Orbitrap Velos

mass spectrometer (Thermo Scientific, Bremen, Germany) coupled to a nano electrospray ionization (nanoESI) source. Filter samples were extracted using a protocol described by Kourtchev et al. (2014). Briefly, the filters undergone three successive extractions in methanol for 15 mins in an ultrasonic bath of slurry ice. The extracts obtained from each extraction were combined and filtered through 0.4 μm and 0.2 μm pore size PTFE filters. The filtered solution was then evaporated under a gentle nitrogen flow until the final volume reached 200 μL before being analyzed with a chip-based nanoESI source (Triversa

NanoMate®, Advion) equipped with a HD ESI A chip (nozzles with 5.5 μm internal diameter, Advion) and operating in negative ion mode with a voltage of −1.40 kV and gas pressure of 0.8 psi, and coupled to a LTQ Velos Orbitrap (Thermo Fisher Scientic) mass spectrometer (MS) with a resolution of 100,000 at $m/z$ 400 and a typical mass accuracy within ± 2 ppm. Data were acquired in full scan in the $m/z$ ranges 50-500 and 150-1000 using the deprotonated ions of the following common background contaminants as lock-masses C$_2$H$_2$O$_2$, C$_{16}$H$_{32}$O$_2$, C$_{18}$H$_{34}$O$_2$, C$_{18}$H$_{36}$O$_2$. Data treatment was performed using a

protocol described in detail by Zielinski et al. (2018). The proprietary software Xcalibur 2.1 was used to assign possible chemical formulas to each signal, considering a maximum allowed mass error of ± 4 ppm and the following parameters: #C 1-100, #H 1-200, #N 0-5, #O 0-50, #S 0-2, #$^{13}$C 0-1, and #$^{34}$S 0-1. Formula lists were further processed using a Mathematica code written in-house that performs internal calibration, noise removal, blank subtraction, and filters formula based on additional atomic constraints (0.3 ≤ H/C ≤ 2.5, O/C ≤ 2, N/C ≤ 1.3, S/C ≤ 0.2,), the nitrogen rule, and isotopic filtering ($^{13}$C/$^{12}$C≤

0.011 and $^{34}$S/$^{32}$S≤ 0.045) (Zielinski et al., 2018). Multiple formula assignments for each peak within instrumental mass accuracy were allowed.

In order to maintain a constant pressure in the CESAM chamber despite sampling by the various instruments, a dry synthetic air (mixture of N$_2$/O$_2$) was continuously injected into the chamber. As a result, all the data presented in this work were corrected for dilution to account for changing of concentrations due to chemistry only. Furthermore, all aerosol data were also corrected

for particle wall losses. This correction took into account the diameter of the particles, and was performed using the model



reported in Lai and Nazaroff, (2000) (friction velocity u* = 3.7 cm s$^{-1}$, from Lamkaddam et al. (2017)). The corrections for particle wall loss were found to be negligeable (less than 5 %).

## 2.2 Chemicals

α-phellandrene was purchased from Sigma-Aldrich (purity > 95 %) and β-phellandrene from Chem Cruz (purity > 96 %). To minimize the presence of volatile impurities, a purification stage was carried out using a vacuum line prior to injecting each of the two VOCs into the chamber. Nitrate radicals were generated *in situ* from the thermal decomposition of $N_2O_5$ following reaction (R9).

$N_2O_5 + M \rightarrow NO_3 + NO_2 + M$ (R9)

$N_2O_5$ was synthesized in a vacuum line following a protocol detailed in Picquet-Varrault et al. (2009). The synthesis involved

two steps: first, the reaction between $O_3$ and $NO_2$ results in the formation of $NO_3$ and then the $NO_3$ reacts with $NO_2$ to produce $N_2O_5$, following reactions (R10) and (R11), respectively.

$O_3 + NO_2 \rightarrow NO_3 + O_2$ (R10)

$NO_3 + NO_2 + M \rightarrow N_2O_5 + M$ (R11)

The resulting dinitrogen pentoxide is then trapped in a vial at 193 K and undergoes purification by pumping. The $N_2O_5$ crystals

can then be stored in a freezer (−18 °C) for several weeks before use.

All experiments were conducted at room temperature and atmospheric pressure, in dry synthetic air, with 80 % of $N_2$ produced from liquid nitrogen boiloff (Messer, purity > 99.995 %, $H_2O$ < 5ppm) and 20 % of $O_2$ from cylinders (Air Liquide, Alphagaz 1, purity > 99.995 %, $H_2O$ < 5ppm).

## 2.3 Experimental protocol

The BVOC was first introduced into the CESAM chamber and was then kept in dark for approximately 1 h to assess any wall loss. No significant wall loss was observed. Following this, $N_2O_5$ was introduced into the chamber using a slow continuous injection method. The glass trap containing frozen $N_2O_5$ crystals was placed in a liquid nitrogen/ethanol bath. $N_2O_5$ was introduced into the chamber by the flow of air over the crystals. The rate of $N_2O_5$ injection was optimized by varying the temperature of the cold bath and the air flow rate in order to allow a progressive and complete consumption of the monoterpene.

## 2.4 Data analyses


The gas phase composition, including reactants and oxidation products, was continuously monitored using multiple analytical techniques as described previously. ONs yields were calculated for both the gas and particulate phases. When using FTIR to quantify the oxidation products, the total organic nitrate yields in the gas phase were determined by plotting their molecular concentration as a function of the reacted BVOC concentration. The slope corresponds to the total ONs yield. Uncertainty on

this yield was calculated as twice the standard deviation on the slope, to which was added the uncertainties on the IBIs of organic nitrates and BVOC. For the particulate phase, the total organic nitrate yield was calculated from the sampled filters





after extraction, by taking the final concentration of ONs and dividing it by the concentration of the reacted BVOC for each experiment. Uncertainty on this yield was calculated as the sum of the relative uncertainties on the IBIs of organic nitrates and BVOC and the uncertainty in the spectra analysis.

ACSM was also used to quantify the total particulate organic nitrate (pON) mass through the fragmentation ratio of the $NO_2^+$ ($m/z$ 46) and $NO^+$ ($m/z$ 30) ions. The methodology used for this quantification has been described in detail elsewhere (Farmer et al., 2010; Kiendler-Scharr et al., 2016; Xu et al., 2015), and is primarily applied to AMS (Aerosol Mass Spectrometer) data. In this work, we aimed to examine if this method can be used for ACSM data (which has lower resolution compared to AMS, with possible fragment artifacts at m/z 30 and 46). The quantification approach is only briefly summarized here. First, the

fraction of organic nitrates ($p_{OrgNO_3frac}$) relative to the mass concentration of the total measured nitrate functionality ($NO_3^-$) is calculated based on $NO_2^+/NO^+$ ratio using the equation (1).

$$p_{OrgNO_3frac} = \frac{\left(1 + R_{OrgNO_3}\right) \times (R_{measured} - R_{calib})}{(1 + R_{measured}) \times (R_{OrgNO_3} - R_{calib})} \tag{1}$$

Where $R_{measured}$ represents the measured $NO_2^+/NO^+$ ratio ($m/z$ 46/30); $R_{calib}$ is the $NO_2^+/NO^+$ ratio associated with inorganic nitrates, determined using pure $NH_4NO_3$ particles during the ACSM calibration (0.7 for our instrument); and $R_{OrgNO_3}$ is the

$NO_2^+/NO^+$ ratio for pure pON. Literature data derived from experiments involving the oxidation of BVOC by $NO_3$ for $R_{OrgNO_3}$ range between 0.1-0.2. For our calculations, we used the value of 0.1, with an estimated uncertainty of 20 % (Bruns et al., 2010; Kiendler-Scharr et al., 2016).

Then, the mass concentration of the nitrate functionality associated with an organic compound ($p_{OrgNO_3mass}$) is calculated using the Equation (2).

$$p_{OrgNO_3mass} = p_{OrgNO_3frac} \times NO_3^- \tag{2}$$

Where $NO_3^-$ represents the total mass concentration of the nitrate functionality.

Finally, the mass concentration of the organic nitrates in the aerosol phase ($p_{ON}$) can be estimated by assuming a mean molecular weight, representative of the various organic nitrates detected in the aerosol phase (see section Results), using the equation (3).

$$p_{ON} = \frac{p_{OrgNO_3mass}}{MW_{NO_3}} \times MW_{pON} \tag{3}$$

Where $MW_{pON}$ and $MW_{NO_3}$ are the molecular weights of the organic nitrates and the nitrate group (62 g mol$^{-1}$) respectively.

The SOA yield for each experiment was calculated by considering each data point during the BVOC decay and continuing after the total consumption of the BVOC, providing both time-dependent and overall SOA yields. Seed particles were not added during the experiments in order to measure SOA yields in conditions of low aerosol content. The SOA yield is defined

as the ratio of the produced SOA mass concentration, $M_0$, to the reacted BVOC mass concentration, $\Delta$BVOC. The uncertainty on $Y_{SOA}$ was estimated as twice the standard deviation of the slope. The SOA yields were plotted as a function of the aerosol mass concentration and fitted by a two-product model parametrization described in equation (4) (Odum et al. 1996).



$$Y_{SOA} = M_0 \left[ \frac{\alpha_1 K_{p,1}}{1 + K_{p,1} M_0} + \frac{\alpha_2 K_{p,2}}{1 + K_{p,2} M_0} \right] \tag{4}$$

Where $\alpha_1$, $\alpha_2$, $K_{p,1}$ and $K_{p,2}$ are respectively the stoichiometric factors and the partitioning coefficients between the gas and

particulate phase (in $m^3\ \mu g^{-1}$) of the hypothetical products.

**2.5 Numerical simulations**

Numerical simulations were performed to identify the dominant reaction pathways for the peroxy radicals under our experimental conditions. Simulations were performed using FACSIMILE software (Curtis, 1980) and oxidation schemes provided by the Master Chemical Mechanism, MCM v3.3.1 (Saunders et al., 2003). As the chemistry of α- and β-phellandrenes

is not available in the MCM, we used the oxidation scheme of limonene, whom chemical structure is very similar to those of phellandrenes. Only the rate constant of the BVOC + $NO_3$ reaction was changed by using the following rate constants: ($3.90 \pm 0.62) \times 10^{-11}$ and $(6.6 \pm 1.0) \times 10^{-12}$ $cm^3$ $molecule^{-1}$ $s^{-1}$ for α- and β-phellandrenes, respectively (Harb et al., 2024). As $NO_3$ and $N_2O_5$ concentrations were below the detection limits, it was not possible to use them as input data for the simulations. So, we used the phellandrene decay rate to constrain the $N_2O_5$ injection rate and consequently the $NO_3$ and $N_2O_5$ concentrations.

**3. Results and discussion**

A total of 8 experiments were carried out for α-phellandrene and 6 experiments for β-phellandrene. Experimental conditions as well as measured total organic nitrates and SOA yields are presented in Table 1. Figure 1 shows an example of the time-series of reactants and products, corrected for dilution, for the experiment conducted on 03/17/2021 for α-phellandrene. The period corresponding to $N_2O_5$ injection is indicated by the yellow hatched area. The concentration of $N_2O_5$ remains below the

detection limit of the FTIR spectrometer (about 5 ppb, for 5 mins of integration time) until the BVOC is completely consumed. It should be noticed that the addition of $N_2O_5$ results in the formation of significant quantities of $HNO_3$ and $NO_2$ due to $N_2O_5$ hydrolysis on surfaces (chamber walls, lines) and decomposition, respectively. Upon complete consumption of α-phellandrene, large quantities of organic nitrates and SOA are formed. In Figure 1, it is shown that, starting with an initial mixing ratio of 75 ppb of α-phellandrene, the formation of up to 20 ppb of total ONs in the gas phase (monitored by FTIR) is observed (Fig. 1.a),

along with approximately 100 µg m$^{-3}$ of particulate organic nitrates (pON, calculated using Eq. 1 to 3 with MW$_{pON}$ = 384.5 g mol$^{-1}$ - refer to Sec. 4.2), and 120 µg m$^{-3}$ of SOA (Fig. 1.b). The aerosol size distribution is also presented (Fig. 1.c), revealing that the particles have mean diameters ranging from 300 to 400 nm. The time profiles of gas phase products detected by the PTR-ToF-MS and the ToF-CIMS are presented in Fig. S1 and Fig. S2 of the Supplement. These profiles will be further analyzed and discussed.

**Table 1: Experimental conditions, total ONs and SOA yields measured for experiments on the oxidation of α- and β-phellandrenes by NO₃ radicals. Missing data are shown by a dash. For molar and mass yields of ONs in the aerosol phase (Y_ONp), LD means that**





the signal was close to the detection limit and could not be quantified. For SOA yields, NA indicates that SOA size distribution exceeded the SMPS size range

| Date | $[BVOC]_i$ (ppb) | Oxidation time (min) | $Y_{ONg}$ molar | $Y_{ONp, FTIR}$ molar | $Y_{ON(p+g)}$ molar | $Y_{ONp, FTIR}$ mass | $Y_{ONp, ACSM}$ mass | $Y_{SOA}$ mass | $\left(\dfrac{Y_{ONp, FTIR}}{Y_{SOA}}\right)$ | $\left(\dfrac{Y_{ONp, ACSM}}{Y_{SOA}}\right)$ |
|---|---|---|---|---|---|---|---|---|---|---|
| α-phellandrene | | | | | | | | | | |
| 03/15/2021 | 290 | 60 | 0.38 ± 0.21 | 0.06 ± 0.02 | 0.44 ± 0.41 | 0.16 ± 0.06 | 0.11 ± 0.02 | NA | - | - |
| 03/16/2021 | 280 | 80 | 0.42 ± 0.18 | 0.06 ± 0.02 | 0.48 ± 0.35 | 0.18 ± 0.06 | 0.11 ± 0.02 | 0.31 ± 0.02 | 0.58 ± 0.22 | 0.35 ±0.09 |
| 03/17/2021 | 75 | 40 | 0.35 ± 0.20 | LD | - | LD | 0.13 ± 0.03 | 0.36 ± 0.04 | - | 0.36 ± 0.11 |
| 03/18/2021 | 50 | 18 | 0.30 ± 0.19 | LD | - | LD | 0.11 ± 0.02 | 0.38 ± 0.09 | - | 0.29 ± 0.13 |
| 03/19/2021 | 70 | 65 | 0.32 ± 0.28 | LD | - | LD | 0.18 ± 0.04 | 0.38 ± 0.05 | - | 0.47 ± 0.16 |
| 06/02/2021 | 250 | 30 | - | 0.03 ± 0.01 | - | 0.08 ± 0.03 | 0.10 ± 0.02 | NA | - | - |
| 06/03/2021 | 480 | 60 | 0.39 ± 0.19 | 0.05 ± 0.02 | 0.44 ± 0.43 | 0.14 ± 0.07 | - | NA | - | - |
| 01/19/2021 | 80 | 60 | - | - | - | - | - | 0.26 ± 0.02 | - | - |
| β-phellandrene | | | | | | | | | | |
| 03/22/2021 | 295 | 100 | 0.48 ± 0.21 | 0.07 ± 0.03 | 0.55 ± 0.49 | 0.20 ± 0.09 | 0.22 ± 0.04 | 0.50 ± 0.01 | 0.40 ± 0.21 | 0.44 ± 0.10 |
| 03/23/2021 | 270 | 95 | 0.51 ± 0.23 | 0.07 ± 0.04 | 0.58 ± 0.56 | 0.20 ± 0.10 | 0.23 ± 0.05 | 0.63 ± 0.02 | 0.32 ± 0.17 | 0.37 ± 0.08 |
| 03/24/2021 | 205 | 70 | 0.53 ± 0.24 | 0.05 ± 0.02 | 0.58 ± 0.56 | 0.14 ± 0.07 | 0.25 ± 0.05 | 0.62 ± 0.02 | 0.22 ± 0.12 | 0.40 ± 0.09 |
| 03/25/2021 | 65 | 65 | 0.52 | 0.04 | 0.56 | 0.13 | 0.27 | 0.67 | 0.19 | 0.40 |



| | | | ± 0.30 | ± 0.02 | ± 0.56 | ± 0.06 | ± 0.05 | ± 0.03 | ± 0.09 | ± 0.10 |
|---|---|---|---|---|---|---|---|---|---|---|
| 03/26/2021 | 67 | 70 | 0.52 ± 0.29 | - | - | - | - | 0.61 ± 0.03 | - | - |
| 01/20/2022 | 62 | 50 | - | - | - | - | - | 0.42 ± 0.01 | - | - |



**Figure 1: Time series of reactants and products during a typical experiment of NO₃-initiated oxidation of α-phellandrene (17/03/2021). (a) Gaseous species mixing ratios, (b) Aerosol mass concentration and chemical composition (with Org. is the total mass concentration of organics, NO₃⁻ is the total nitrate functionality mass concentration, pOrgNO₃ is the mass concentration of the nitrate group within particulate organic nitrates-pON), and (c) Aerosol size distribution and mass concentration. All plots are corrected from dilution. The N₂O₅ injection period is shown by the yellow hatched area.**



## 3.1 SOA yields

In Figure 2, the SOA yields ($Y_{SOA}$) are plotted against the total organic aerosol mass concentration ($M_o$) for $\alpha$- and $\beta$-phellandrenes. Notably, the experimental data points obtained from multiple experiments demonstrate good agreement. The experimental data were fitted using a two-product parametrization (see Eq. 4). The parameters obtained for $\alpha$-phellandrene are: $\alpha_1 = 3.6 \times 10^{-1}$; $K_{p,1} = 4.4 \times 10^{-2}$ m$^3$ µg$^{-1}$ and $\alpha_2 < 1.0 \times 10^{-3}$; $K_{p,2} < 1 \times 10^{-2}$ m$^3$ µg$^{-1}$. The second class of products was observed to have a negligible impact on the SOA formation, allowing a good fit using only one class of products. For $\beta$-phellandrene also, one class of products allows to reproduce the experimental data points using the following parameters: $\alpha_1 = 5.4 \times 10^{-1}$; $K_{p,1} = 4.0 \times 10^{-2}$ m$^3$ µg$^{-1}$. For both compounds, the parametrization succeeds to reproduce the experimental data points over the entire range of aerosol content. These results show that both monoterpenes are efficient SOA precursors with final yields up to 35 % and 60 % for $\alpha$-phellandrene and $\beta$-phellandrene, respectively. According to the Odum parametrization, the partitioning coefficient of the products responsible for the SOA formation ($K_{p,1}$) is similar for both monoterpenes, but their formation yield ($\alpha_1$) is higher for $\beta$-phellandrene. To our knowledge, this study provides the first determination of SOA yields for the NO$_3$-initiated oxidation of $\alpha$- and $\beta$-phellandrenes.

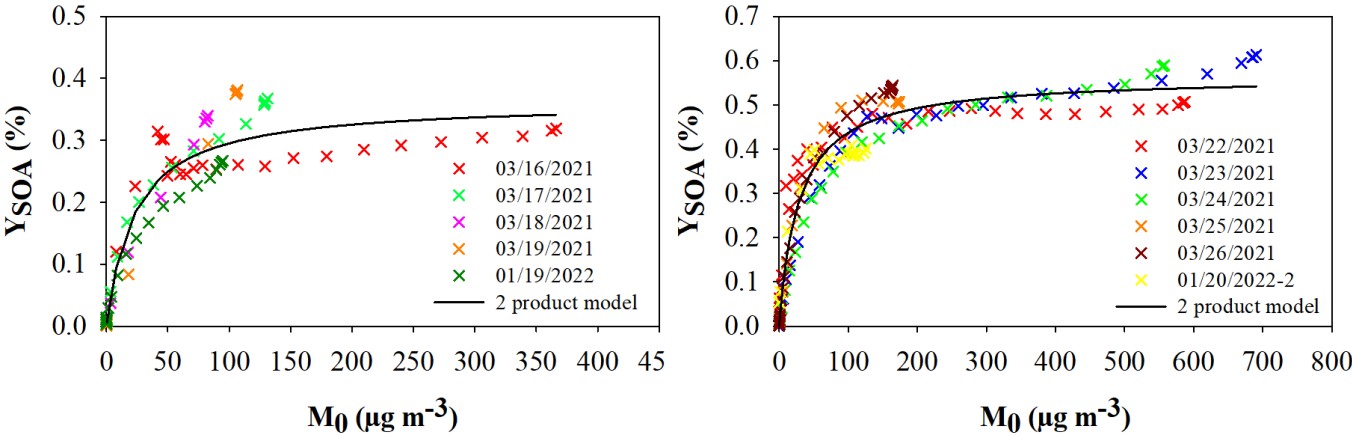

**Figure 2: SOA yields as a function of the SOA mass concentration measured for $\alpha$-phellandrene (left plot) and $\beta$-phellandrene (right plot). Data points from different experiments are shown with different colors and Odum parametrization plots correspond to the black solid curves.**

These SOA yields were also compared to those available in the literature for other monoterpenes having similar chemical structures. Here, we only considered studies performed using the same experimental conditions (no seeds, slow and continuous injection of N$_2$O$_5$, …) to enable a relevant comparison. Hence, as discussed in the introduction, SOA yields measured for a single monoterpene have been observed to significantly vary from a study to another one and this may be explained by differences in the RO$_2$ chemical regimes. Odum plots for $\alpha$- and $\beta$-phellandrenes (this study) are compared to those obtained for $\alpha$- and $\gamma$-terpinenes (Fouqueau et al., 2020) and for terpinolene (Fouqueau et al., 2022) in Fig. S3. These studies were also performed using the CESAM chamber and the same experimental protocol. It can be observed that $\beta$-phellandrene and




terpinolene are the monoterpenes which generate the most SOA with maximum yields around 50-60 %. In a lesser extent, α-phellandrene and γ-terpinene produce SOA with yields up to 30-40 %. Finally, α-terpinene has much lower SOA yield which does not exceed 2 %. It is interesting to note that the monoterpenes with the highest SOA yields are the ones which exhibit an exocyclic double bond. It can also be noticed that these five monoterpenes exhibit different aerosol production dynamics: for α- and β-phellandrenes, the SOA yields increase rapidly and reach a maximum for an aerosol content around 400 µg m$^{-3}$ indicating the formation of low volatility products ($K_p \sim 4 \times 10^{-2}$ m$^3$ µg$^{-1}$). On the contrary, the increase of the SOA yield is more progressive for γ-terpinene and terpinolene indicating the formation of another class of products having higher volatility ($K_p \sim$ 3-6 $\times 10^{-3}$ m$^3$ µg$^{-1}$) which significantly contributes to the SOA formation. For α-terpinene, the stochiometric factors α were observed to be very small (~ 0.01) leading to very low SOA yields (Fouqueau et al., 2020).

**3.2 Organic nitrate yields**

The total content of particulate organic nitrates (pON) was quantified using two methods as described previously: (i) by sampling aerosols on filters, followed by off-line FTIR spectroscopy, and (ii) in real-time, by using the ACSM. ON yields in the aerosol phase ($Y_{ONp}$) are presented in Table 1. The molar $Y_{ONp}$ ranges from 3 to 7 % for both compounds. For experiments with low BVOC mixing ratios, the concentrations of ONs in the particulate phase are close to the detection limit of the FTIR technique, and their quantification was not possible. To assess the contribution of organic nitrates to the SOA, a comparison of the $Y_{ONp}$ with the SOA yields, both expressed in mass, has been performed. The molar $Y_{ONp}$ values were converted into mass yields using $MW_{pON} = 384$ g mol$^{-1}$, which is the average of the molecular weights of two types of ONs: a monomer $C_{10}H_{15}NO_4$ (MW = 213 g mol$^{-1}$) and a dimer $C_{20}H_{32}N_2O_{16}$ (MW = 556 g mol$^{-1}$). These two compounds were detected in both gas and aerosol phases with intense signals. This assumption on the molecular weight introduces an uncertainty in the estimation of the mass yield of organic nitrates. The ON mass yields calculated from off-line FTIR analyses ($Y_{ONp, FTIR}$)$_{mass}$ and on-line ACSM measurements ($Y_{ONp, ACSM}$)$_{mass}$ are presented in Table 1. A reasonable agreement is observed between the two techniques considering the uncertainties and assumptions in the ACSM data treatment as well as the overall uncertainty on the filter sampling, extraction, and FTIR analysis. To our knowledge, this is the first comparison of ON quantification in the aerosol phase using ACSM and off-line FTIR. It can be concluded that the ACSM can be used to estimate the total ON concentration in the aerosol phase even though it is associated with a large uncertainty due to assumptions made for the data treatment. The main advantage of ACSM in comparison to off-line FTIR are that first, it does not require sample preparation and analysis, and then, it provides time-resolved measurements.

Since the individual quantification of oxidation products was not performed, this approach allows to estimate the fraction of organic nitrates in the particulate matter. The mass yields of ONs in the aerosol phase were observed to range from 8 to 18 % for α-phellandrene and from 13 to 27 % for β-phellandrene, depending on the experiment and the technique used. To estimate the contribution of ONs to the aerosol phase, the ratios $Y_{ONp,mass}/Y_{SOA,mass}$ were calculated and are provided in Table 1. These



results indicate that for both monoterpenes, ONs are major constituents of the SOA, and their contribution can reach up to 50 %.

The formation of gas-phase ONs was also monitored using *in situ* FTIR spectroscopy. The total ONs yield in the gas phase ($Y_{ONg}$) was determined for both monoterpenes by plotting the mixing ratio of ONs against the mixing ratio of the reacted monoterpene (see Fig. 3). Yields measured for each single experiment are provided in Table 1. A good agreement among the

different experiments is observed, with the experimental points well fitted by the linear regression, indicating that the $Y_{ONg}$ remains constant throughout the experiments. Moreover, the plot reveals a non-zero slope at the origin, suggesting that organic nitrates are primary products and can undergo further reactions to produce secondary organic nitrates.

Previous studies (Picquet-Varrault et al., 2020; Suarez-Bertoa et al., 2012) have shown that organic nitrates may undergo losses on the stainless-steel chamber walls. The loss rate for various multifunctional ONs has been determined, with values ranging

from 0.5 to $2 \times 10^{-5}$ $s^{-1}$. Under our experimental conditions, where the $Y_{ONg}$ values were calculated over a relatively short time (less than 1 h), the wall losses are estimated to be less than 10 %. This estimation is confirmed by the good linearity observed in the plot. The molar yields ($Y_{ONg}$) obtained for α- and β-phellandrenes were found to be 40 ±12 % and 49 ±16 %. These findings show that organic nitrates are major products of the NO₃-initiated oxidation of these monoterpenes, as observed for other terpenes. For example, Fouqueau et al. (2020a, 2022) found $Y_{ONg}$ of 43 ±10 % for α-terpinene, 47 ±10 % for γ-terpinene

and for terpinolene, and 43 ±10 % for β-caryophyllene. Similarly, $Y_{ONg}$ obtained for limonene ranged from 30 % to 72 % (Hallquist et al., 1999; Spittler et al., 2006; Fry et al., 2011, 2014), while for β-pinene it varied between 40 % and 74 % (Hallquist et al., 1999; Fry et al., 2014; Boyd et al., 2015).

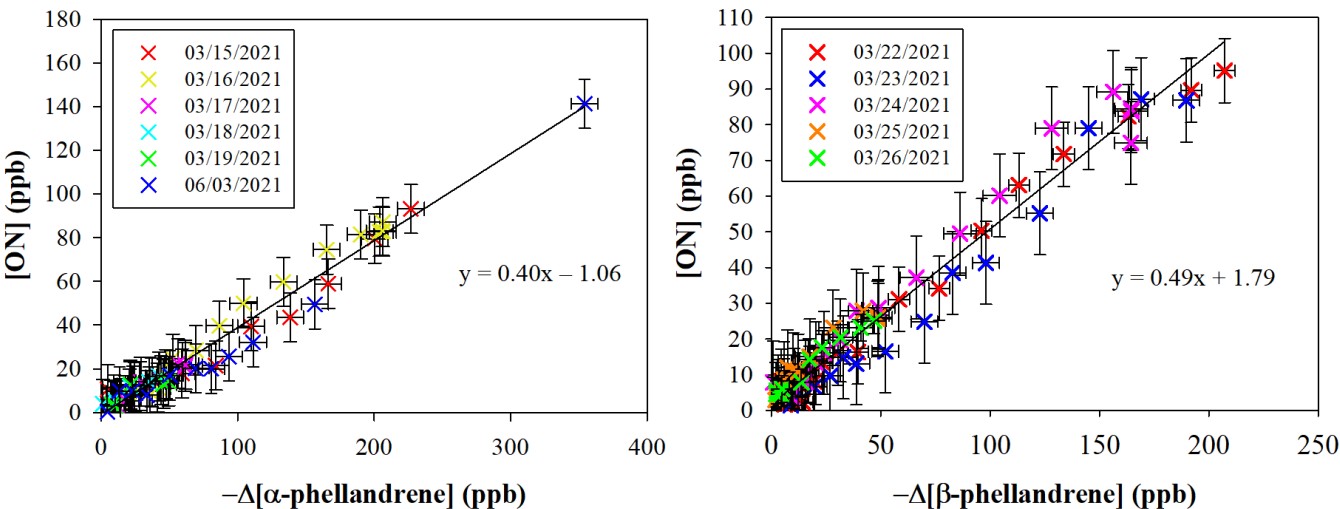

**Figure 3: Formation of gas-phase organic nitrates as a function of the loss of α-phellandrene (left) and β-phellandrene (right).**



### 3.3 Gas-phase products identification and formation pathways

In this section, the gas-phase products measured by PTR-ToF-MS are discussed. HOMs products measured by ToF-CIMS are discussed in a dedicated section. Table S1 provides a summary of the $m/z$ and molecular formulas of the products detected using PTR-ToF-MS in $H_3O^+$ and $NO^+$ ionization modes. Time profiles of these products (illustrated in Fig. S1 for α-phellandrene) were used to determine whether they are first or second-generation products. Several products have been detected in both ionization modes, reinforcing confidence in their identification. For example, for α-phellandrene, the product $C_{10}H_{15}NO_4$ (MW = 213 g mol$^{-1}$) has been detected at m/z 214 as [M+H]$^+$ in $H_3O^+$ mode and at m/z 243 as [M+NO]$^+$ in $NO^+$ mode.

Mechanisms have been proposed to explain the formation of these products. $NO_3$ reacts with α- and β-phellandrenes through addition onto one of the two C=C bonds (H-abstraction being likely a negligible pathway) leading to the formation of four different nitrooxy-alkyl radicals. Due to the presence of conjugated double bonds in phellandrenes, electron delocalization can lead to the formation of isomers (see Fig. 4). Although the favored position for $NO_3$ addition is the one resulting in the most substituted radical (Kerdouci et al., 2010), all alkyl radicals have been considered here. It is worth mentioning that most products derived from these various alkyl radicals are isomers that cannot be differentiated from each other with the analytical measurement techniques employed here. To facilitate the reading of the mechanism, pathways for only two radicals will be presented in the following.

**Figure 4: Different possible alkyl radicals resulting from NO₃ addition onto one of the two carbon double bonds of α-phellandrene**

The oxidation scheme of α-phellandrene is presented in Fig. 5 while the one of β-phellandrene is available in Fig. S4. Considering that only few differences were observed in the oxidation products for both monoterpenes, we will present in detail the oxidation scheme of α-phellandrene and will only discuss the differences observed for β-phellandrene. First generation



products which were detected are indicated in blue, while second generation products are highlighted in red. Following the addition of $NO_3$, the nitrooxy-alkyl radicals likely react rapidly with $O_2$ leading to the formation of peroxy radicals, $RO_2$ (reaction 2 in Fig. 5). These $RO_2$ radicals can further evolve following several reaction pathways described in previously (R1-5), through various radical-radical reactions ($RO_2 + RO_2$, $RO_2 + NO_3$ and $RO_2 + HO_2$). As mentioned in the introduction, the

$RO_2$ fate significantly affects the product distribution and the SOA yields (Bates et al., 2022; Day et al., 2022). Numerical simulations were therefore performed to identify the dominant $RO_2$ reaction pathways under our experimental conditions. Results are shown in Fig. 6 for the experiment of 01/19/2022. It can be observed that the $RO_2$ self-reactions represent more than 99 % of the total $RO_2$ consumption during the oxidation of the BVOC. This can be explained by the fact that $N_2O_5$ was introduced continuously in limited amount during the experiment, so that $NO_3$ was a limiting reactant. Once the BVOC is

consumed, the contribution of $RO_2 + NO_3$ reactions increases slightly and can represent up to a few % (approximately 7 %) of the $RO_2$ loss, which suggests that these reactions should not be neglected, in particular when considering second-generation products. These numerical simulations also confirm that $NO_3$ mixing ratios are very low and do not exceed few ppt. In the following discussion, only $RO_2 + RO_2$ and to a lesser extent, $RO_2 + NO_3$ will be considered, whereas $RO_2 + HO_2$ will be considered negligible as it represents less than 1 % with $HO_2$ concentrations below few $10^4$ molecule $cm^{-3}$.

Alkoxy radicals (RO) can be formed via the $RO_2 + RO_2$ (reaction 3 in Fig. 5) and the $RO_2 + NO_3$ (Reaction 3' in Fig. 5) pathways. The $RO_2$ self-reaction may also form an hydroxynitrate (see reaction R1). However, no hydroxynitrate (MW = 215 g $mol^{-1}$) was observed, suggesting that this reaction is a negligible pathway. This reaction involves an H-atom transfer and requires at least one hydrogen atom linked to the carbon carrying the peroxy radical (i.e. not a tertiary peroxy radical). Considering that in our system, the most favorable peroxy radical is expected to be a tertiary one, this reaction would be indeed

negligible. The hydroxynitrate was however observed for other monoterpenes having similar chemical structures, such as γ-terpinene and terpinolene (Fouqueau et al., 2020). Alkoxy radicals can undergo various pathways following reactions 4, 5, 6 and 7 in Fig. 5: (i) Through reaction 4, they can decompose by cleaving the $C(ONO_2)−CH(O^•)$ bond, resulting in a ring opening and the formation of a dicarbonyl compound with MW = 168 g $mol^{-1}$. This product was detected at m/z 169 in $H_3O^+$ ionization mode and at m/z 168 in $NO^+$ mode. (ii) The ring opening can also occur through a scission on the other side of the alkoxy

group, leading to the formation of an alkyl radical (reaction 5). This latter can further evolve to the formation of trifunctional diketonitrate product with a molecular weight of 229 g $mol^{-1}$, detected with a weak signal at m/z 230 in $H_3O^+$ ionization mode and at m/z 259 in $NO^+$ mode. (iii) Alkoxy radicals can also react with $O_2$ via reaction 6, leading to the formation of a ketonitrate with MW = 213 g $mol^{-1}$, detected at m/z 214 in $H_3O^+$ ionization mode and at m/z 243 in $NO^+$ mode. (iiii) Finally, the presence of an adjacent double bond to the alkoxy group can allow for an epoxidation reaction (reaction 7 in Fig. 5) (Vereecken et al.,

2021; Wang et al., 2013), resulting in the formation of a nitrated epoxy alkyl radical. This radical can further evolve leading to the formation of a multifunctional epoxide having the same molecular weight (229 g $mol^{-1}$) as the previously observed diketonitrate formed via reaction 5. As mentioned before, isomers cannot be distinguished by PTR-ToF-MS measurements. In addition, a compound with a molecular weight of 152 g $mol^{-1}$ was detected at m/z 153 using $H_3O^+$ ionization mode. Previous studies conducted in the CESAM chamber on α-terpinene, γ-terpinene, and terpinolene (Fouqueau et al., 2022, 2020) have



also reported this mass, and it was attributed to an epoxide formed through the loss of the $NO_2$ group from a nitrooxy alkyl radical. However, this pathway is usually considered to be negligible in the presence of $O_2$. A possible explanation is that this signal results from a fragmentation process occurring in the drift tube of the PTR-ToF-MS. Other peaks were detected (e.g. m/z 149, 153) and corresponding raw formula were proposed but no formation pathways could be established to explain their formation. These peaks may also result from fragmentation processes.

First generation products that still possess a double bond can react further with $NO_3$ leading to second generation products colored in red in Fig. 5. This is supported by the time-dependent decrease of signals corresponding to primary products with molecular weights of 168 g mol$^{-1}$ and 213 g mol$^{-1}$ (see Fig. S2), as well as by the detection of secondary products with MW of 72 g mol$^{-1}$, 128 g mol$^{-1}$, and 245 g mol$^{-1}$. The compounds with MW of 72 and 128 g mol$^{-1}$ correspond to second generation dicarbonyl products formed by decomposition of alkoxy radicals (reactions 4-3 and 4-4 in Fig. 5), themselves resulting from

the oxidation of dicarbonyl compounds (MW = 168 g mol$^{-1}$). These radicals can further evolve via reactions 2-3' and 2-4' to form highly functionalized products (tricarbonyl nitrates) with a MW of 245 g mol$^{-1}$. These compounds were detected at m/z 245 in $NO^+$ ionization mode. Additionally, the oxidation of the ketonitrate (MW = 213 g mol$^{-1}$) can also lead to the formation of a compound with the same MW of = 245 g mol$^{-1}$.

For β-phellandrene, the same first-generation products were detected but an additional product having the MW = 138 g mol$^{-1}$

was observed. This product corresponds to an unsaturated carbonyl compound which is formed from an alkoxy decomposition, as shown in Fig. S4 (reaction 4-1). Few differences were also observed for the second-generation products.







**Figure 5: Proposed mechanism for the oxidation of α-phellandrene by NO₃ radical. First-generation products are colored in blue, and second-generation ones are colored in red.**


**Figure 6: Results of the numerical simulation for the experiment of 01/19/2022. a) Modeled concentrations of: α-phellandrene, RO₂, NO₃ and HO₂. b) Contributions of RO₂, NO₃ and HO₂ to RO₂ total consumption.**



**3.4 HOMs identification and formation pathways**

In addition to products detected by PTR-ToF-MS, gas-phase HOMs were monitored using ToF-CIMS. It was observed that

HOMs are formed in the very first stage of the oxidation (see time profiles in Fig. S2) suggesting that these compounds are formed through autoxidation processes rather than being second generation products. Similar HOMs were detected for both monoterpenes, with only minor differences. Two types of HOMs, having distinct ranges of m/z ratio, were observed: HOM monomers with *m/z* between 300 and 400 and HOM dimers with *m/z* between 500 and 650 (see Fig. S5 and Fig. S6). For α-phellandrene, a total of 22 HOM compounds were identified, including 8 monomers ($C_9$ and $C_{10}$) and 14 dimers ($C_{17}$ and $C_{20}$).

For β-phellandrene, 24 HOMs were identified, including 9 monomers and 15 dimers. All detected products contain at least six O atoms and one or two N atoms. Surprisingly, no HOM peroxy radicals were observed in this study. Identified products were categorized into families, grouping compounds that share the same number of C-, N-, and H-atoms, but different numbers of O-atoms. In total, we identified 3 monomer families ($C_9H_{14}N_2O_{9-10}$; $C_{10}H_{15}NO_{8-11}$ and $C_{10}H_{17}NO_{8-10}$) and 2 dimer families ($C_{17}H_{26}N_2O_{12-18}$ and $C_{20}H_{32}N_2O_{10-18}$). The carbon number offers valuable insight into potential fragmentation resulting from

C-C bond cleavage, particularly for any carbon number that is not equal to 10 (for monomers) or 20 (for dimers). The hydrogen number provides information regarding terminal functional groups and bimolecular reactions involved in HOM formation. The nitrogen number can serve as an indicator of second-generation products or $NO_2 + RO_2$ reactions. The oxygen number provides an indication on how much autoxidation process occurred.

The most abundant HOM monomers detected in the gas-phase for both monoterpenes are presented in Table 3. They are $N_1$-

$C_{10}$ stable closed-shell products: $C_{10}H_{15}NO_x$ (x = 8-11), which correspond to carbonyl compounds, and $C_{10}H_{17}NO_x$ (x = 8-10), which correspond to hydroxyl or hydroperoxide compounds. The time series of these HOMs showed a pattern of first-generation products. Their formation can be explained by the autoxidation of peroxy radicals which is described in Fig. 7. The first peroxy radicals being formed by the reaction of $NO_3$ with phellandrene (followed by reaction with $O_2$) are nitrooxy-peroxy ($NO_3$-$RO_2$) radicals $C_{10}H_{16}NO_5{}^{\bullet}$. These radicals can then undergo an autoxidation through H-shift and $O_2$ addition, leading to

the formation of new nitrooxy-peroxy radicals ($C_{10}H_{16}NO_7{}^{\bullet}$) which contain an additional –OOH functional group (Fig. 7-a). This H-shift can proceed several times resulting in a progressive series of compounds with the general formula $C_{10}H_{16}NO_{2n+1}{}^{\bullet}$ with an odd number of O atoms (see example in Fig. 7-a). This unimolecular pathway was shown to be favored by the presence of a C=C double bond in a $NO_3$-$RO_2$ radical (Møller et al., 2020; Vereecken and Nozière, 2020). Furthermore, nitrooxy-peroxy radicals can also form nitrooxy-alkoxy ($NO_3$-$RO$) radicals via bimolecular reactions involving $RO_2 + RO_2$ and $RO_2 + NO_3$.

The resulting $NO_3$-$RO$ radical can also undergo H-migration and $O_2$ addition, leading to the formation of new nitrooxy-peroxy radicals ($C_{10}H_{16}NO_6{}^{\bullet}$) which contain an additional –OH functional group (see Fig. 7-a). This "mixed" alkoxy-peroxy autoxidation pathway has been reported in the literature (e.g. (Shen et al., 2021)) and results in a series of $C_{10}H_{16}NO_{2n}{}^{\bullet}$ with an even number of O atoms. In addition, as discussed previously, a nitrated-epoxy-peroxy radical ($C_{10}H_{16}NO_6{}^{\bullet}$) can be formed via reaction 7 in Fig. 5. In this case, if we consider a comparable autoxidation pathway to that previously described for the

nitrooxy-peroxy radical $C_{10}H_{16}NO_5{}^{\bullet}$, it will result in a sequential series of $C_{10}H_{16}NO_{2n}{}^{\bullet}$ radicals with an even number of O



atoms. On the other hand, the "mixed" alkoxy-peroxy pathway leads to a progressive series of $C_{10}H_{16}NO_{2n+1}^{\bullet}$ radicals with an odd number of O atoms. This mechanism is illustrated in Fig. 7-b.

The radical chain termination of $C_{10}H_{16}NO_{2n+1}^{\bullet}$ and $C_{10}H_{16}NO_{2n}^{\bullet}$ can occur through unimolecular termination channels, such as OH loss following H-abstraction from a carbon with a –OOH group attached, or through bimolecular reactions such as $RO_2$
+ $RO_2$. These termination reactions lead to the formation of closed-shell carbonyl-nitrates, hydroxynitrates and hydroperoxynitrates. One should note that hydroxynitrates formed from $C_{10}H_{16}NO_x^{\bullet}$ and hydroperoxynitrates formed from $C_{10}H_{16}NO_{x-1}^{\bullet}$ have the same molecular formula and cannot be distinguished based on mass spectra analysis. Among the $C_{10}$-HOM monomers, $C_{10}H_{15}NO_8$ and $C_{10}H_{17}NO_8$ were identified as the most abundant. Notably, the signal of $C_{10}H_{15}NO_8$ which corresponds to a carbonylnitrate was found to be higher than that of $C_{10}H_{17}NO_8$ which corresponds to a hydroxy/hydroperoxy-
nitrate. This has also been observed in the chemical systems involving β-pinene + $NO_3$ (Dam et al., 2022; Shen et al., 2021) and limonene + $NO_3$ (Guo et al., 2022). Considering that the bimolecular reaction (R1) ($RO_2 + RO_2 \rightarrow ROH + R(O)$) leads to equal amounts of carbonyl and hydroxyl HOMs, this result suggests that additional reaction pathways produce carbonyl HOMs. Among the most probable pathways, the reaction of alkoxy radicals with $O_2$ forms carbonyls and $HO_2$ (Atkinson et al., 2008). In addition, recent studies have proposed a unimolecular termination reaction of $RO_2$ via H-shifts (Crounse et al., 2013; Mentel
et al., 2015; Vereecken and Nozière, 2020) leading to higher abundance of carbonyl HOMs compared to hydroxy/hydroperoxy HOMs.

Finally, we also identified a family of $C_9$-HOMs ($C_9H_{14}N_2O_{9-10}$) which were more abundant for β-phellandrene than for α-phellandrene. These products can be formed by the cleavage of an exocyclic or endocyclic C-C bond followed by autoxidation. For β-phellandrene, a first-generation $C_9H_{14}N_2O_9$ compound and a second-generation $C_9H_{14}N_2O_{10}$ compound were detected
but we are not able to propose detailed formation mechanisms at this stage.

**Table 2: $N_1$-$C_{10}$ termination products, including carbonyl compounds (M-17), hydroxyl compounds (M-15) and hydroperoxide compounds (M+1) formed by the oxidation of α- and β-phellandrenes by $NO_3$ radicals. The corresponding $RO_2$ radicals ($C_{10}H_{16}NO_x^{\bullet}$) were not detected but proposed from the detection of termination products.**

| Peroxy radical (M) | Carbonylnitrate (M-17) | Hydroxynitrate (M-15) | Hydroperoxynitrate (M+1) |
|---|---|---|---|
| $C_{10}H_{16}NO_8^{\bullet}$ | / | / | $C_{10}H_{17}NO_8$ |
| $C_{10}H_{16}NO_9^{\bullet}$ | $C_{10}H_{15}NO_8$ | $C_{10}H_{17}NO_8$ | $C_{10}H_{17}NO_9$ |
| $C_{10}H_{16}NO_{10}^{\bullet}$ | $C_{10}H_{15}NO_9$ | $C_{10}H_{17}NO_9$ | $C_{10}H_{17}NO_{10}$ |
| $C_{10}H_{16}NO_{11}^{\bullet}$ | *$C_{10}H_{15}NO_{10}$ | $C_{10}H_{17}NO_{10}$ | / |
| $C_{10}H_{16}NO_{12}^{\bullet}$ | $C_{10}H_{15}NO_{11}$ | / | / |

*: Not detected for α-phellandrene

Dimers are formed by $RO_2$ accretion reactions (R5) ($RO_2 + R^{'}O_2 \rightarrow ROOR^{'} + O_2$) (Berndt et al., 2018b; Dam et al., 2022;
Shen et al., 2022; Zhao et al., 2018). In this study, two types of accretion products were identified, $C_{17}H_{26}N_2O_{12-18}$ and



$C_{20}H_{32}N_2O_{10-18}$, with the $C_{20}$ dimers being the most abundant. For both monoterpenes, the temporal profiles of these HOM dimers showed a pattern of first-generation products, with very intense peaks present immediately after the initiation of the phellandrene oxidation, followed by a rapid decrease (see Fig. S2). Remarkably, following the drop of the dimer signals to zero, the number of particles formed, and their total mass reached their peak values. This observation strongly supports the significant role of dimers in particle formation and growth. Within the $C_{20}$ family, $C_{20}H_{32}N_2O_{16}$ was found to be the most abundant compound and exhibits the highest signal among all detected HOMs (monomers and dimers). Considering that the $RO_2$ self-reactions can involve two identical $RO_2$ radicals or two different types of $RO_2$, $C_{20}H_{32}N_2O_{16}$ can result from cross-reactions such as $C_{10}H_{16}NO_9{}^{\bullet}$ + $C_{10}H_{16}NO_9{}^{\bullet}$, $C_{10}H_{16}NO_{10}{}^{\bullet}$ + $C_{10}H_{16}NO_8{}^{\bullet}$, or $C_{10}H_{16}NO_{11}{}^{\bullet}$ + $C_{10}H_{16}NO_7{}^{\bullet}$. Regarding $C_{17}H_{26}N_2O_{12-16}$ dimers, their formation may involve the reaction between a $C_7$-$RO_2$ radical and a $C_{10}$-$RO_2$ radical. It should be noted that the generation of a $C_7$-$RO_2$ radical can occur through the alkoxy carbon chain decomposition and fragmentation of a $C_{10}$ monomer. It is expected that this fragmentation also leads to the formation of acetone. This was confirmed by the detection of acetone using PTR-ToF-MS. In conclusion, HOM monomers and dimers resulting from autooxidation processes have been detected with very similar patterns for both monoterpenes. The concomitant disappearance of dimers with the aerosol formation suggests that dimers play a significant role in the particle formation. However, we cannot exclude that HOM monomer also contribute to it. No significant differences were observed in the gas-phase HOMs that could explain the differences observed in the SOA formation for the two monoterpenes.



**Figure 7: Illustrative scheme of H-shifts for peroxy and alkoxy radicals formed by the reaction α-phellandrene + NO₃. a) for unsaturated peroxy radicals; b) for epoxy-peroxy radicals.**

### 3.5 Particulate-phase products identification

The chemical composition of the aerosol phase was investigated by filter sampling and analysis by Orbitrap-MS. Almost all HOM monomers and dimers which were detected in the gas-phase, including $C_{10}H_{15}NO_{7-11}$, $C_{10}H_{17}NO_{7-10}$, $C_{17}H_{26}N_2O_{10-17}$ and $C_{20}H_{32}N_2O_{10-18}$, were also observed in the aerosol phase. The fact that the same HOM monomers were detected in both phases indicates that monomers also contribute to the SOA formation. As for the gas phase, the most abundant monomers and dimers



were found to be $C_{10}H_{15-17}NO_8$ and $C_{20}H_{32}N_2O_{16}$. New products, which were not observed in the gas-phase, were detected in the aerosol phase with high abundance, such as $C_8H_{12}O_2$, $C_9H_{14}O_{4-5}$ and $C_{10}H_{18}N_2O_{10-11}$. Remarkably, within the dimers, new families of products with N=3,4,5 were identified. In particular, $C_{20}H_{33}N_3O_{14-19}$ were detected with very high abundance. If these products are formed in the gas phase, they should be detected by the ToF-CIMS, unless they rapidly partition into the condensed phase due to their low volatility. Another possible explanation is that these products are formed through chemical

processes occurring in the particulate phase.

The products were classified into two categories according to their raw formulas: CHO and CHON. The relative abundance of each group was determined based on the intensity of their chromatographic peaks (%). It is important to keep in mind that different organic species may have different signal responses in mass spectrometry, but due to the lack of standards, we assumed that all species have the same response. Figure 8 compares, for α- and β-phellandrenes, the relative abundance of

CHO and CHON products, their classification into monomers and dimers, and the distribution of CHON compounds as a function of the number of N-atoms. It can be deduced that for both monoterpenes, CHON is the most abundant category, with a relative abundance higher than 80 %, while CHO compounds contribute to less than 20 %. These findings confirm that organic nitrates are major constituents of SOA resulting from the oxidation of phellandrenes by $NO_3$. Moreover, within the CHON group, dimers (10<C≤30) represent more than 80%. Among the monomers, $C_{10}$ compounds are the most abundant,

while $C_{20}$ compounds are the most abundant dimers (see Fig. S7). The distribution of CHON compounds as a function of the number of N shows the formation of products with N>2 (N=3,4,5) which were not detected in the gas-phase (see Fig. 8). Also, some differences are observed in their distribution between the two monoterpenes. The contributions of 4-N and 5-N products are higher for β-phellandrene than for α-phellandrene, whereas the contribution of 3-N compounds is similar. In total, CHON compounds with N>2 represent up to 63 % for β-phellandrene and 53 % for α-phellandrene. However, 2-N products are more

abundant for α-phellandrene. Given that the addition of a nitro group leads to a decrease of the volatility (Kroll and Seinfeld, 2008), these differences in N number may explain the differences observed in the SOA yields. Additionally, differences were observed in the distribution of products as a function of their C number (see Fig. S7). A non-negligible contribution of compounds with a carbon number lower than 9 in the particulate phase was observed for α-phellandrene, suggesting more fragmentation of the oxidation products for α-phellandrene than for β-phellandrene. The formation of these compounds is

expected to result from the cleavage of a C-C bond adjacent to the alkoxy group. This can also explain why β-phellandrene is a more efficient SOA precursor than α-phellandrene.



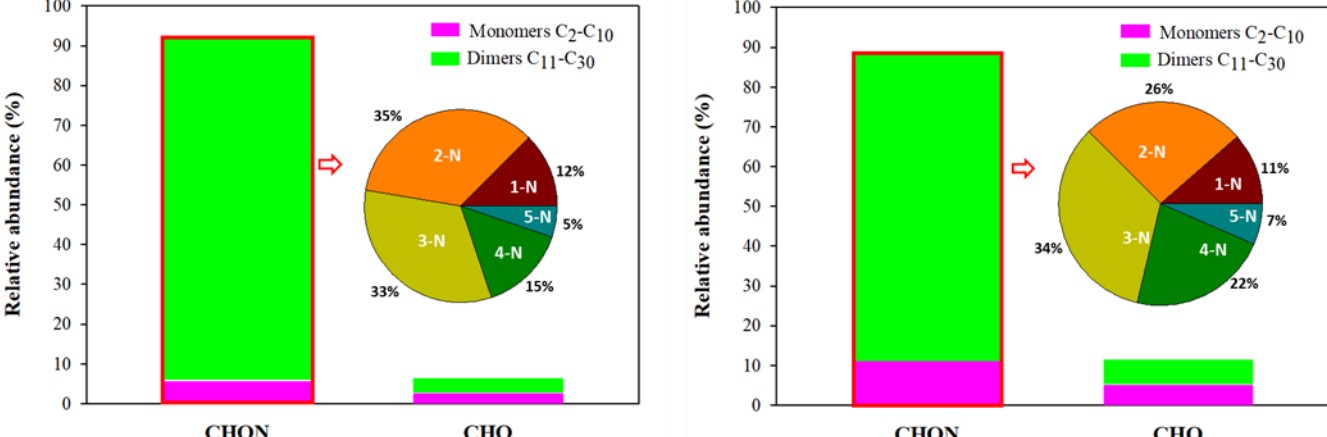

**Figure 8: The relative abundance of CHON and CHO compounds in the particulate phase, their classification into monomers and dimers, and the distribution of CHON compounds as a function of the number of N-atoms. Left: for α-phellandrene; right: for β-phellandrene.**

## 4. Conclusion

This study has provided in-depth investigation of the $NO_3$-initiated oxidation of α- and β-phellandrenes thanks to experiments in the CESAM simulation chamber and by combining a wide variety of analytical techniques (FTIR, PTR-ToF-MS, ACSM, ToF-CIMS, Orbitrap, SMPS). SOA yields were measured, and oxidation products, including HOMs, were investigated in the gas and aerosol phases. Numerical simulations were also performed to investigate the dominant chemical regimes for $RO_2$ radicals. To our knowledge, this study is the first mechanistic study on the reactions of α- and β-phellandrenes with $NO_3$ radical.

In this study, we have shown that α- and β-phellandrenes are efficient SOA precursors with yields reaching up to 35 % and 60 %, respectively. Considering that experiments were performed using the same protocol, we could deduce that β-phellandrene generates significantly more SOA than α-phellandrene. We also showed that these reactions produce large amounts of organic nitrates in both the gas and aerosol phases, similarly to what was observed for other monoterpenes. Total molar yields of organic nitrates (gas + aerosol) were found to range between 40 and 60 % and these compounds were shown to be major constituents of the SOA, with mass contribution up to 50 %. This was confirmed by Orbitrap analyses which have shown that relative abundance (in signal intensity) of CHON compounds is higher than 80 %.

To gain deeper insight into the mechanisms, chemical analyses of the products in both gaseous and particulate phases were performed using various mass spectrometry techniques. For both monoterpenes, several types of first-generation products were detected, including carbonyl nitrates, dicarbonyl nitrates and dicarbonyls. Second-generation products which result from the addition of $NO_3$ radical onto the second C=C bond, were also detected. They are of two types: i) tricarbonyl nitrates which are low volatility products, and ii) products resulting from fragmentation such as dicarbonyls. We could also detect a number of



gas-phase HOM monomers and dimers and highlight the significance of the autooxidation processes for these monoterpenes. Numerical simulations have shown that, under our experimental conditions, $RO_2$ radicals evolve mainly by cross reactions ($RO_2 + RO_2$). This is consistent with the detection of gas-phase accretion products (dimers). Detailed chemical mechanisms have been proposed accordingly. Given that gas-phase products were similar for the two monoterpenes, they cannot explain the differences in SOA yields. However, some differences in the aerosol phase chemical composition have been observed:

products with high number of N atoms (4-N and 5-N) which could result from condensed phase chemistry, were observed to be more abundant for β-phellandrene than for α-phellandrene. In addition, products having less than 10 C-atoms were more abundant for α-phellandrene, suggesting more fragmentation of the oxidation products of α-phellandrene than for β-phellandrene. These observations may explain why β-phellandrene is a more efficient SOA precursor than α-phellandrene. Deeper investigation of these processes is needed to better understand differences in the SOA yields observed between α- and

β-phellandrenes.

## Data availability

The data that support the findings of this study are openly available in EUROCHAMP-2020 data centre at https://www.eurochamp.org/.

## Author contributions

BPV and MCi coordinated the research. SH, BPV, MCi designed the experiments in the simulation chamber. SH performed the experiments with the technical support from BPV, MCi, MCa, EP, AB, SA (ToF-CIMS), CC (ToF-CIMS), VM (ToF-CIMS). SH also performed data treatment and interpretation with MCi and BPV. SA performed data treatment for the ToF-CIMS. Orbitrap analyses were conduted by FB. CG provided the expertise in Orbitrap-MS analyses. SH drafted the initial manuscript. All authors made contributions to this work and approved the final version of the manuscript.

## Competing interests

C. Cantrell is co-organizer of the special issue "Atmospheric Chemistry of the Suburban Forest – multiplatform observational campaign of the chemistry and physics of mixed urban and biogenic emissions (ACP/AMT inter-journal SI)". The remaining authors declare no competing interests.

## Acknowledgments

The authors gratefully acknowledge CNRS-INSU for supporting CESAM platform as a component of the ACTRIS Research Instructure. This work has benefited from the support of the research infrastructure ACTRIS-FR and the European



Commission, Horizon 2020 Research Infrastructures (EUROCHAMP-2020, grant n° 730997). This work was also supported by the French Environment and Energy Management Agency (ADEME), by Paris Region in the framework of call for projects from DIM Qi2, by the French National Research Agency (ACROSS project, ANR-17-MPGA-0002) and by the French

National program LEFE of the CNRS/INSU. Orbitrap analyses were supported by a BP Next Generation fellowship awarded by the Yusuf Hamied Department of Chemistry at the University of Cambridge to CG. The authors also thank AERIS (https://www.aeris-data.fr/) for curing and distributing the data within EUROCHAMP Data Center. The authors also want to thank ICARE insitute, and especially Wahid Mellouki and Véronique Daële, for the loan of their CI-APi-ToF-MS inlet without which this study would have not been possible.

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
