# Peer review of "HOMs and SOA formation from the oxidation of $\alpha$ - and $\beta$ -phellandrenes by NO3 radicals"

_EGUsphere, 2024_

## Referee Comment (RC1)

**Overall comments:**

The study by Harb et al. investigates the formation of secondary organic aerosol (SOA) from oxidation of α- and β-phellandrene by nitrate radicals (NO3). This oxidation pathway denotes an important oxidation and SOA formation pathway during nighttime. The experiments are carried out in the laboratory and a suite of instruments is deployed to characterize both particle and gas phase during their experiments. The authors find relatively high SOA yields for both precursors studied and considerable fraction of condensed phase organic nitrates. Reaction mechanisms are proposed and cross reactions of RO2 radicals (RO2+RO2) are found to be a dominant for the conditions of their laboratory experiments. Overall, I find this study to be an important contribution to the literature of SOA formation by NO3 oxidation, which compared to oxidation of VOCs by OH or O3 is understudied.

The manuscript is very well written and structured, allowing the reader to follow the analysis of the experimental results and the interpretation thereof. I therefore recommend publication of this manuscript, after the authors have addressed the few specific comments that I list below.

**Specific comments:**

L100-104: More quantitative information should be added to the emission strength of α- and β-phellandrene. Knowing the area covered by the stated Eucalyptus species is good to know, but I am curious how much vapors get emitted into the atmosphere of a given VOC type. How does this compare to other major SOA precursors such as α-pinene? What is the global atmospheric relevance of the systems studied? Please extend the discussion in the manuscript.

L117: It is not obvious to me how the low level of electrostatic charges on the chamber wall leads to longer aerosol lifetime in the chamber? I assume this is due to reduced diffusional loss? Please extend the discussion and provide appropriate references.

L215-219: In Section 2.3, I would appreciate it if the authors could be a bit more quantitative. E.g., What was the air flow rate over the crystals? Over which range was the temperature varied/adjusted? This information could easily be added to the text.

L225: "organic nitrate yield" use the introduced acronym, i.e.: "ON yield"

Table 1:

- Please add information about what the stated values and the range indicated by "±" means to the Table caption. How is this uncertainty calculated?
- Please decide if you want to report your SOA yields as percent values (see my comment below to Fig. 2).

L299-301: "… allowing a good fit using only one class of products." How can I see this from your Fig. 2? Is it possible to add a fit line (e.g., as dashed line) to your Fig. 2 where you only use the term for the first class of products of eq. (4)? To visualize this effect.

L304: Replace "Odum parameterization" with "eq. (4)"

Fig. 2:

- SOA yield is not given in units of %, as suggested by the y-axis. In general, I find it hard that you switch back and forth between yields given in percent vs. fractional values between Table 1, the Figures and the text. I would encourage you to choose e.g. %-values and consistently use these throughout.
- Left-side panel: The x-axis is cut, the upper limit should be "450", not "45".
- Change caption to read on L310: "… colors and Odum parameterization corresponds to the black solid curves (eq. 4)."
- Please use panel labels (a), (b), … for this figure as you do in Fig. 1, for consistency. This also applies to other figures, e.g., Fig. 3 and Fig. 8.

L342-342: "… the total ON concentration in the aerosol phase even though it is associated with large uncertainty due to assumptions made for the data treatment." What are the assumptions and how large

are these uncertainties? These uncertainties are not reflected in the values listed in Table 1 (at least the uncertainties listed there seem small to me), please clarify.

L347-348: Please add uncertainties as e.g. "range from 8 ± xx to 18 ±xx %" and consider reporting yields in units of % also in Table 1. Also, according to your Table 1 (column Y_ONp,ACSM) it appears that the smallest ON mass yield was 10% (06/02/2021), and not 8%.

L350: add "can reach up to approximately 50%." (the largest yield value in Table 1 is 0.47)

L356: "plot reveals a non-zero slope at the origin". It is unclear to me what you mean here?

L361: "to be less than 10%". On L197 you claim less than 5%, please choose appropriately.

L450: Looking at your Fig. S2 it appears that some of the traces slope downwards. On L194 you state that your data is wall loss corrected, so I would have expected the traces in Fig. S2 to stay constant at a certain value, after the initial increase. Why is this not the case?

L464: I cannot find any "Table 3".

L470: Please put comma in front of "which" (almost always)

L477: note correct formatting of parenthesis: "(e.g. Shen et al., 2021)".

L494: Delete "recent"

L508: "Remarkably, following the drop of the dimer signals to zero, the number of particles formed, and their total mass reached their peak values." (and your statement on L518-519) Can you visualize this by adding traces for the particle mass concentration of e.g., your SMPS data to Fig. S2?

L527: Replace "which" by "that" (also on L546)

L529: Would it be more appropriate to say "…also contribute to SOA formation and growth."? Otherwise, how do you disentangle formation and growth from your experimental data?

L573: "… constitutes of the SOA, …" I feel it would be more appropriate to say "particle phase" here instead of SOA, at least this is how I understand your statement. Might be good to clarify your statement here.

---

## Author Response (AR1)

RC1

**Overall comments:**

**The study by Harb et al. investigates the formation of secondary organic aerosol (SOA) from oxidation of α- and β-phellandrene by nitrate radicals (NO3). This oxidation pathway denotes an important oxidation and SOA formation pathway during nighttime. The experiments are carried out in the laboratory and a suite of instruments is deployed to characterize both particle and gas phase during their experiments. The authors find relatively high SOA yields for both precursors studied and considerable fraction of condensed phase organic nitrates. Reaction mechanisms are proposed and cross reactions of RO2 radicals (RO2+RO2) are found to be a dominant for the conditions of their laboratory experiments. Overall, I find this study to be an important contribution to the literature of SOA formation by NO3 oxidation, which compared to oxidation of VOCs by OH or O3 is understudied.**
**The manuscript is very well written and structured, allowing the reader to follow the analysis of the experimental results and the interpretation thereof. I therefore recommend publication of this manuscript, after the authors have addressed the few specific comments that I list below.**

We would like to sincerely thank the reviewer for the time spent reviewing our manuscript and for the thoughtful comments provided. We appreciate the positive feedback on the study and the structure of the manuscript. We have carefully considered all of the reviewer's comments and suggestions and have made revisions accordingly.

**Specific comments:**

**L100-104: More quantitative information should be added to the emission strength of α- and β-phellandrene. Knowing the area covered by the stated Eucalyptus species is good to know, but I am curious how much vapors get emitted into the atmosphere of a given VOC type. How does this compare to other major SOA precursors such as α-pinene? What is the global atmospheric relevance of the systems studied? Please extend the discussion in the manuscript**

The paragraph has been modified following the reviewer's recommendation (L100-111): *"Among all emitted monoterpenes, α-phellandrene has been identified as a major component in extracts and emissions from numerous Eucalyptus species (Li, Madden, and Potts 1995; Maghsoodlou et al. 2015; Maleknia, Bell, and Adams 2009; Pavlova et al. 2015; He, Murray, and Lyons 2000), which are the most widely planted hardwood forest trees on the global scale (>20 million ha) (Myburg et al. 2014).* **It can account for up to 21% of the monoterpene emission rate in some Eucalyptus species (He, Murray, and Lyons 2000). To our knowledge, while no global annual emission data are available for α-phellandrene, it is listed among the most frequently emitted monoterpenes in the EMEP/EEA air pollutant inventory Guidebook (2023).** *β-phellandrene is a major contributor to emissions from coniferous trees, such as Scots pine and Norway spruce (Hao et al. 2009; Janson 1992; Joutsensaari et al. 2015; Ylisirniö et al. 2020; Yassaa et al. 2012).* **Yassaa et al. (2012) showed that it can constitute approximately 27 % of the total isoprenoid emissions from Scots pine and 15 % of the monoterpene emissions from Norway spruce. β-phellandrene is also considered as one of the major emitted monoterpenes, with a global annual total emission estimated at 1.5 Tg yr$^{-1}$ (A. B. Guenther et al. 2012b; "EMEP/EEA Air Pollutant Emission Inventory Guidebook" 2023)."**

**L117: It is not obvious to me how the low level of electrostatic charges on the chamber wall leads to longer aerosol lifetime in the chamber? I assume this is due to reduced diffusional loss? Please extend the discussion and provide appropriate references.**

Electrostatic charges contribute to attracting the aerosol particles, which are charged, to the chamber walls. This has been shown in several papers (Wang et al., 2018; Charan et al, 2018). In particular, Charan et al. (2018) studied the charge effect on the rate of particle wall deposition, estimating both the eddy-diffusion coefficient and the average magnitude of the electric field within their chamber. These electrostatic effects were shown to be important in Teflon chambers, thus affecting the aerosol lifetimes. They are shown to be smaller in stainless steel reactors. More information can be found in the book entitled "A practical Guide to atmospheric simulation chambers" (https://doi.org/10.1007/978-3-031-22277-1) (Doussin et al., 2023). This reference has been added in the manuscript (L126).

**L215-219: In Section 2.3, I would appreciate it if the authors could be a bit more quantitative. E.g., What was the air flow rate over the crystals? Over which range was the temperature varied/adjusted? This information could easily be added to the text.**

We thank the reviewer for this suggestion. The optimal conditions for the cold bath temperature were between 203 K and 193 K, and for the flow rate over the $N_2O_5$ crystals, between 0.5 and 3 L min$^{-1}$. This missing information has now been added to the text in line 228 for clarity:

*"The rate of $N_2O_5$ injection was optimized by varying the temperature of the cold bath **(between 203 K and 193 K)** and the air flow rate **(between 0.5 and 3 L min$^{-1}$)** in order to allow a progressive and complete consumption of the monoterpene."*

**L225: "organic nitrate yield" use the introduced acronym, i.e.: "ON yield"**

Thank you for pointing this out. The modification has been made.

Table 1:

**- Please add information about what the stated values and the range indicated by "±" means to the Table caption. How is this uncertainty calculated?**

Thank you for your comment. To address the request, we have updated the table caption by adding *Absolute uncertainties are indicated after "±"*.

All the uncertainties presented are absolute uncertainties (A.U.). The calculation of relative uncertainties (R.U.) is described in Section 2.4.

- For YONg (FTIR), see lines 234-236:

$$Total\ R.U.on\ YONg\ (FTIR) = R.U.on\ the\ slope + R.U.on\ the\ IBI\ of\ ONs + R.U.on\ the\ IBI\ of\ the\ BVOC$$

- For YONp (FTIR), see lines 238-239:

$$Total\ R.U.on\ YONp\ (FTIR) = R.U.in\ the\ spectra\ analysis + R.U.on\ the\ IBI\ of\ ONs + R.U.on\ the\ IBI\ of\ the\ BVOC$$

- For YONp (ACSM), see lines 251-252:  R.U. on YONp (ACSM) = 20 %
- For YSOA, see line 265-266: R.U. on YSOA = R.U. on the slope
- For YON(p+g) molar: Total R.U. ON YON(p+g) was calculated as the sum of the total R.U. on both YONp and YONg
- For YONp/YSOA : Total R.U. on YONp/YSOA was calculated as the sum of the total R.U. on both YONp and YSOA

**- Please decide if you want to report your SOA yields as percent values (see my comment below to Fig. 2).**
The modification has been made. All yields in Table 1 are now reported in % units

**L299-301: "... allowing a good fit using only one class of products." How can I see this from your Fig. 2? Is it possible to add a fit line (e.g., as dashed line) to your Fig. 2 where you only use the term for the first class of products of eq. (4)? To visualize this effect.**
To address the reviewer's suggestion, we have added a dashed fit line to Figure 2, representing a fit using only the first class of products from Eq. (4). As shown in the updated figure, the fit remains unaffected. This visualization helps to clarify the point made in lines 320-321: *"... allowing a good fit using only one class of products."*

**L304: Replace "Odum parameterization" with "eq. (4)"**
The modification has been made. (L324)

Fig. 2:

**- SOA yield is not given in units of %, as suggested by the y-axis. In general, I find it hard that you switch back and forth between yields given in percent vs. fractional values between Table 1, the Figures and the text. I would encourage you to choose e.g. %-values and consistently use these throughout.**
We apologize for the confusion regarding the units of SOA yield in Figure 2. The y-axis in Figure 2 now corresponds to $Y_{SOA}$ in % units. For consistency, all yields in Table 1 are also now reported in % units, as suggested.

**- Left-side panel: The x-axis is cut, the upper limit should be "450", not "45".**
Thank you for pointing this out. The modification has been made

**- Change caption to read on L310: "... colors and Odum parameterization corresponds to the black solid curves (eq. 4)."**
The modification has been made.

**- Please use panel labels (a), (b), ... for this figure as you do in Fig. 1, for consistency.**
The modification has been made

**L342-342: "... the total ON concentration in the aerosol phase even though it is associated with large uncertainty due to assumptions made for the data treatment." What are the assumptions and how large are these uncertainties? These uncertainties are not reflected in the values listed in Table 1 (at least the uncertainties listed there seem small to me), please clarify.**

In fact, we are primarily referring to the ±20% uncertainty mentioned in line 251, which arises from the choice of the $R_{orgNO_3}$ value.

**L347-348: Please add uncertainties as e.g. "range from 8 ± xx to 18 ±xx %" and consider reporting yields in units of % also in Table 1. Also, according to your Table 1 (column Y_ONp,ACSM) it appears that the smallest ON mass yield was 10% (06/02/2021), and not 8%.**

Uncertainties have been added to Lines 367 and 368. Yields in Table 1 have also been updated by using % units. As mentioned in Lines 368-369, the reported range of ON mass yields in the aerosol phase includes data from both techniques (FTIR and ACSM). The value of 8% corresponds to the ON mass yield for the same experiment (now labeled as experiment 6) column $Y_{ONp}$ FTIR mass in Table 1.

**L350: add "can reach up to approximately 50%." (the largest yield value in Table 1 is 0.47)**
This modification has been made (L371).

**L356: "plot reveals a non-zero slope at the origin". It is unclear to me what you mean here?**
In Figure 3, the X-axis represents the amount of reacted MT, while the Y-axis represents the concentration of ONs formed. The slope at the origin is not zero indicating that ONs are produced immediately and proportionally as MT is consumed, consistent with the behavior of first-generation products. If ONs were secondary products, their formation would depend on intermediates or additional reaction steps. In such a scenario, the slope near the origin would approach zero, as no ONs would form until intermediates accumulated. The observed linear relationship, starting with a measurable slope at the origin, supports the conclusion that ONs are primary products.
The sentence has been rephrased in the manuscript to make it clear.

**L361: "to be less than 10%". On L197 you claim less than 5%, please choose appropriately.**
The difference arises because the values refer to different aspects of the study. In Line 205, the "less than 5 %" refers to the particle wall loss, whereas in Line 383, the "less than 10 %" corresponds to the wall losses estimated for gas-phase organic nitrates.

**L450: Looking at your Fig. S2 it appears that some of the traces slope downwards. On L194 you state that your data is wall loss corrected, so I would have expected the traces in Fig. S2 to stay constant at a certain value, after the initial increase. Why is this not the case?**
The wall loss correction was applied to the particle phase, as mentioned in Lines 202-204, and no correction for wall loss was performed for the gas phase products. Wall losses for organic nitrates were estimated to be less than 10 % as previously mentioned. As indicated in Lines 294-295, the concentration of $N_2O_5$ remains below the detection limit of the FTIR spectrometer until the BVOC is completely consumed. After this point, the $N_2O_5$ concentration increases, reaches a maximum, and then decreases (Fig. 1.a). Since $N_2O_5$ is the source of $NO_3$, we still have $N_2O_5$ in the chamber after the oxidation of the BVOC. This could lead to further oxidation of these compounds and may explain the observed downward slope in some of the traces in Figure S2.
**L464: I cannot find any "Table 3".**
We thank the reviewer for pointing this out. This was indeed a mistake, and the reference to "Table 3" has been corrected to "Table 2" in the text.

L470: **Please put comma in front of "which"** (almost always)
This modification has been made.

**L477: note correct formatting of parenthesis: "(e.g. Shen et al., 2021)".**
This modification has been made (L504).

**L494: Delete "recent"**
This modification has been made.

**L508: "Remarkably, following the drop of the dimer signals to zero, the number of particles formed, and their total mass reached their peak values." (and your statement on L518-519) Can you visualize this by adding traces for the particle mass concentration of e.g., your SMPS data to Fig. S2?**
In response to the reviewer's suggestion, we have added the particle mass concentration, based on the SMPS data, to Figure S2.

**L527: Replace "which" by "that" (also on L546)**

This modification has been made (L567 and L547).

**L529: Would it be more appropriate to say "…also contribute to SOA formation and growth."? Other-wise, how do you disentangle formation and growth from your experimental data?**

Thank you for pointing this out.
Based on our experimental data, we observe that the temporal profile of dimers shows a rapid increase once BVOC oxidation begins (Figure S2), followed by a rapid decrease in their signals, coinciding with an increase in particle size and number/mass concentration. Additionally, dimers are also detected in the particle phase. This suggests their participation in the formation and/or the growth of SOA but we agree with the reviewer that we cannot distinguish their contribution in both processes. The sentence has been modified accordingly by replacing "formation and growth" by "formation and/or growth". The final sentence is *"this rapid decrease of dimers coincides with the particle formation, suggesting that HOMs dimers also contribute to SOA formation and/or growth".*

**L573: "… constitutes of the SOA, …" I feel it would be more appropriate to say "particle phase" here instead of SOA, at least this is how I understand your statement. Might be good to clarify your statement here.**
This modification has been made (L595).
In our experiment, SOA and the particulate phase are effectively the same, as no seed particles were used. Additionally, ACSM data confirmed that the particles consist predominantly of organic constituents.

**References:**

Charan, S.M., Kong, W., Flagan, R.C., Seinfeld, J.H., 2018. Effect of particle charge on aerosol dynamics in Teflon environmental chambers. Aerosol Science and Technology 52, 854–871. https://doi.org/10.1080/02786826.2018.1474167

Doussin, J.-F., Fuchs, H., Kiendler-Scharr, A., Seakins, P., Wenger, J. (Eds.), 2023. A Practical Guide to Atmospheric Simulation Chambers. Springer Nature. https://doi.org/10.1007/978-3-031-22277-1

**RC2**

This study by Harb et al. examines the gas-phase oxidation of phellandrene isomers by NO3 radicals. They conducted gas-phase measurements via FTIR, a combination of PTR/NO+ TOF-MS and NO3-CI-TOF-MS, and particle measurements via offline Orbitrap, ACSM and SMPS. Overall, their work reports the SOA yields for these reactions, along with qualitative particle phase composition measurements (orbitrap). From their online measurements of the gas-phase components they propose in-depth oxidation mechanisms.

**Overall comment:**

**With the manuscript in its current form, I do not recommend this work for publication. This is primarily due to the over-use of the gas-phase data to create detailed mechanisms, when no speciated structural information or quantitation was obtained. Setting the lack of structural information aside, beyond the obvious limitation of the method used (PTR/NO+ TOF-MS) that isomers cannot be resolved, at best this method offers complex detection of these multifunctional species with high LODs. In general, traditional PTR/NO+ methods are not appropriate for the detection of multi-functional (especially organic nitrate) compounds due to fragmentation, unless a pre-separation method is used to quantitatively interpret the complex product ion distributions. I read the predecessor paper (Duncianu et al., 2017), and while the authors were able to demonstrate the formation of molecular ion adducts - the spectra of these synthesized standards were still complex (3 - 5 ions formed per compound), and generally dominated by fragmentation. I do not think it is appropriate to apply this method to a complex system like the oxidation of phellandrenes without supporting, speciated, methods for product identification and quantitation - especially if the aim is to create a detailed oxidation mechanism.**

We thank the reviewer for his comments. However, we would like to clarify some points raised in his comments.

The mechanism proposed based on PTR-MS measurements aims to propose possible formation pathways for the detected products and molecular formulas, including different isomers, as multiple chemical structures are possible for the same m/z. We never claimed to provide any quantification or to identify the major pathways in the mechanism and we fully agree on the fact that this mechanism may not be complete as it is only based on PTR-MS measurements and some products may not have been detected. Many papers in the literature use the same approach (e.g. Fayad et al., 2021; Fouqueau et al., 2022, 2020) and propose mechanisms based on mass spectrometry techniques without isomer resolution. From the reviewer comment, we understand that the way the mechanism was presented was not clear, and to clarify it, the title of the Figure 5 has been modified by replacing *"Proposed mechanism for the oxidation of α-phellandrene by NO₃ radical. First-generation products are colored in blue, and second-generation ones are colored in red."* by *"Mechanism proposed to explain the formation of first-generation products (colored in blue) and second-generation products (colored in red) detected by PTR-ToF-MS for the reaction of α-phellandrene+NO₃."* Additionally, the text has been modified accordingly (L23-24, L410-413, L604-605). The title of figure S2 was also updated.

Concerning the method used to identify the products by PTR-MS, we do not agree on the fact that it is not suitable for the detection of organic nitrates. "In general", PTR-MS is not suitable for the detection of organic nitrates. But here, we operated the instrument with a much lower electric field in the drift tube (E/N ≈ 40 Td) than normally used (E/N ≈ 130 Td) to limit fragmentation. In the

previous paper from Duncianu et al., 2017 which present the optimization of this method, the mass spectra of alkyl, hydroxy- and carbonyl nitrate standards were measured using both NO⁺ and H₃O⁺ ionization modes, with low E/N (see for example the mass spectra of nitrooxypropanol (left) and nitrooxy-acetone (right) in the figure below). We agree that several peaks are observed but for nitrooxy-acetone, the peak at m/z 120 (M+1) is the most intense one and for both compounds, the adduct formation is the most intense peak in NO⁺ mode.

[Figure]

To our knowledge, all ionization methods, even the "soft" methods, including the conventional PTR-MS with H₃O⁺, lead to some fragmentation. In the figure below (from Kilpinen et al., 2012), is shown an example of fragmentation patterns for different VOCs. Many of them undergo fragmentation and exhibit much more than 3-4 peaks. In spite of this, mass spectrometry methods are powerful techniques to measure VOCs and are widely used by the atmospheric community (e.g. Kim et al., 2010; Li et al., 2020; Maji et al., 2020), in particular in ambient air where mixtures are much more complex than in simulation chambers.

**Table 1.** Fragmentation pattern in the PTR-MS.

| Compound | Ion masses (relative abundance) |
|---|---|
| Acetone | 59 (100) |
| Propanal | 59 (100), 31 (18) |
| Hexanal | 55 (100), 83 (74), 101 (4), 53 (2) |
| (E)-2-Hexenal | 57 (100), 99 (23), 81 (21), 43 (6) |
| Heptanal | 55 (100), 97 (57), 69 (9), 115 (4), 53 (2) |
| Octanal | 69 (100), 41 (47), 111 (26), 55 (11), 71 (7), 129 (6), 67 (2) |
| (E)-2-Octenal | 109 (100), 57 (53), 127 (33), 67 (28), 59 (4), 83 (2) |
| Nonanal | 69 (100), 83 (33), 55 (32), 57 (24), 143 (9), 125 (7), 71 (6), 67 (4) |
| Decanal | 83 (100), 55 (92), 69 (22), 97 (20), 157 (13), 81 (8), 139 (2), 53 (2) |
| Undecanal | 55 (100), 43 (66), 97 (51), 83 (38), 69 (34), 171 (19), 111 (8), 81 (3), 53 (2) |
| Sulcatone | 109 (100), 127 (25), 69 (3), 67 (1) |
| Geranyl acetone[a] | 177 (100), 109 (31), 113 (30), 121 (26), 69 (21), 81 (21), 195 (19), 137 (15), 139 (15), 99 (10), 107 (9), 85 (8), 127 (7), 83 (4) |
| Benzaldehyde | 107 (100), 79 (11) |
| Benzylalcohol | 91 (100), 79 (25) |
| Limonene[b] | 81 (100), 137 (27), 95 (10), 93 (1), 107 (1), 121 (1) |

But the most important point is that we used this dual-mode approach using both NO⁺ and H₃O⁺ ionization modes to identify the products. The simultaneous detection of a product in both ionization modes strengthens the confidence in the assigned molecular formulas and minimizes the risk of attributing fragments to incorrect parent molecules.

**Specific comments:**

**1. In section 2.4 the gas- and particle-phase concentrations of organic nitrates (ONs) are discussed. Was FTIR used for both the gas- and particle-phase measurement? The authors mention collecting particles followed by extraction, but the detection method is not clear. Below they also mention online pON from the ACSM. Also, in line 223-224 referring to these as "molecular concentrations" is mis-leading since this is a bulk measurement.**

We thank the reviewer for this comment. FTIR was used for both gas and particulate phase measurements: in-situ long-path FTIR was used for gas-phase measurements (see description in lines 149-160) and off-line FTIR was used for the aerosol phase. Indeed, as explained in the section 2.1 of the manuscript (lines 179-183), off-line FTIR was used to measure the total organic nitrates (ONs) in the particulate phase after having performed extraction in $CCl_4$. For more clarity, we added a reminder in line 235 on the detection method. The ACSM was also used to determine the total particulate ONs concentration, as mentioned in line 240.

We understand that the term "molecular concentrations" could be misleading in this context, as it might be interpreted as referring to individual molecular-level measurements (e.g., a direct molecular count). To improve clarity and better reflect the nature of the measurements conducted using FTIR, we propose replacing "molecular concentrations" with "measured concentrations," which is more appropriate in this context. This modification has been made in the manuscript (L233). We thank the reviewer for this relevant comment.

**2. In section 2.5, the authors mention that the phellandrene oxidation mechanism is not included in the MCM and so they used the limonene mechanism as a proxy. Considering the authors claim that small changes in structure can create large differences in SOA yield/chemistry (e.g., large difference in measured SOA yield between alpha- and beta-phellandrene reported here) the authors need to provide more justification that the use of the limonene mechanism is valid here. For example, does the temporal profile of the decay of precursor and growth of first-generation oxidation products' align? Are the rate constants comparable? Furthermore, has the limonene + NO3 mechanism currently included in the MCM been validated?**

We maintain that small changes in the VOC structure may significantly affect its reactivity, and this was confirmed by numerous studies in the literature but also by the rate constants for limonene and phellandrenes, which are significantly different. Therefore, we agree that the limonene oxidation scheme cannot be used to describe/explain in detail the formation of individual products or SOA from phellandrene oxidation. However, this was not our objective. **We used the numerical simulations only to estimate the $NO_3$, $HO_2$ and total $RO_2$ concentrations so we can then calculate the rates of the $RO_2+RO_2$, $RO_2+NO_3$ and $RO_2+HO_2$ reactions. To do so, we have of course replaced the rate constant of limonene by the one of α- or β-phellandrene, as it was indicated in the manuscript in section 2.5, L276-277.** As $NO_3$ and $N_2O_5$ concentrations were below the detection limits, we used the phellandrene decay rate (and the rate constant phellandrene + $NO_3$ which is known) to constrain the $N_2O_5$ injection rate and consequently the $NO_3$ and $N_2O_5$ concentrations. With this method, we also assume that $RO_2$ concentrations are well simulated as $RO_2$ formation rate is directly correlated with the VOC oxidation rate and slight changes in $RO_2$ chemistry won't significantly affect their total concentration.

Finally, to calculate the rate of the reactions $RO_2+RO_2$, $RO_2+NO_3$ and $RO_2+HO_2$, we used the generic rate constants provided by the MCM for peroxy radicals: $k(RO_2+NO_3) = 2.3\times10^{-12}$ $cm^3$ $molecule^{-1}$ $s^{-1}$; $k(RO_2+HO_2) = 2.2\times10^{-11}$ $cm^3$ $molecule^{-1}$ $s^{-1}$; For $RO_2$ self-reaction, the rate constant differs with the type of $RO_2$ (primary, secondary, tertiary). For limonene and phellandrenes, $NO_3$-

oxidation is expected to form mainly the most-substituted tertiary alkyl radical, leading to tertiary peroxy radicals. Therefore, we used, as proposed by MCM, the generic rate constants for tertiary peroxy radicals: $k(RO_2+RO_2) = 9.2 \times 10^{-14}$ cm$^3$ molecule$^{-1}$ s$^{-1}$;

The use of generic rate constants generates uncertainty on the results, but this is the best that can be done given the lack of experimental data for $RO_2$ radicals. Also, the objective here is not to determine precisely the reaction rates, but rather to estimate the order of magnitudes.

All this information has been added in the manuscript in section 2.5 (L279-289).

**3. In Table 1, the "Date" column can be eliminated and replaced with "experiment 1" etc. This format should be carried out to other figures/discussion (e.g., Figure 2, legend).**

Following the reviewer's comment, the "Date" column was removed and replaced with "Experiment [number]." This format has been applied consistently to other figures and throughout the discussion.

**4. Section 3.3, line 355 - 357: The off-set at the origin (Figure 3) between measured ON and reacted MT, doesn't this indicate a high background? I'm not sure I understand the statement this the "non-zero slope at the origin suggests these are primary products."**

The offset at the origin is very small and not significant. To demonstrate this, the uncertainty on the ordinate at the origin has been calculated as $2 \times \sigma$ (2 x standard deviation) for the two plots. For $\alpha$-phellandrene, the ordinate at the origin is $1 \pm 2$; for $\beta$-phellandrene, it is $2 \pm 2$. It can be concluded that the ordinate at the origin is not significantly different from zero. This information has been added in Figure 3.

In Figure 3, the X-axis represents the amount of reacted MT, while the Y-axis represents the concentration of ONs formed. The slope at the origin is not zero indicating that ONs are produced immediately and proportionally as MT is consumed, consistent with the behavior of first-generation products. If ONs were secondary products, their formation would depend on intermediates or additional reaction steps. In such a scenario, the slope near the origin would approach zero, as no ONs would form until intermediates accumulated. The observed linear relationship, starting with a measurable slope at the origin, supports the conclusion that ONs are primary products. The sentence has been rephrased in the manuscript to make it more clear (L376-379): *"Moreover, the slop at the origin is not zero indicating that ONs are produced immediately and proportionally as monoterpene is consumed, which is consistent with the behavior of first-generation products. The fact that the slope remains constant during the experiments also indicates if primary ONs are consumed, they will evolve towards the formation of secondary ONs."*

**5. Discussion, lines 395 - 410: Throughout the discussion the authors mention that in their experiments the RO2 fate was dominated by RO2 + RO2 reactions (> 95 %). However, they say that the generation of the closed shell products from the RO2 + RO2 reaction (either self or cross) to form a hydroxy and carbonyl product pair (Russell Mechanism) was negligible. This disagrees with past measurements which have shown branching ratios of averaging around 50/50 between closed shell products and alkoxy radicals, while variable depending on structure the molecular channel is not negligible for these types of RO2 structures (see review by Orlando and Tyndall, 2012). For other systems that also form first generation tertiary RO2 radicals, the closed shell hydroxy nitrate still form through cross reactions in non-negligible quantities (e.g., Claflin and Ziemann, 2018). In this work, the lack of measurement of the hydroxy nitrate, in tandem with the lack of speciated/specific measurements and the complex spectra generated from the method utilized, more so point to the measurement itself not being appropriate to detect these compounds.**

The measurement method used in this study (PTR-MS with two ionization modes: $H_3O^+$ and $NO^+$) was previously employed in other studies conducted by our group under similar experimental conditions (Fouqueau et al., 2020, 2022) on three monoterpenes (terpinolene, α-terpinene, and γ-terpinene) reacting with $NO_3$. These studies demonstrated that hydroxynitrates can indeed be detected, but their formation depends on the specific structure of the monoterpene. Hydroxynitrates (M=215 g.mol$^{-1}$) were detected for γ-terpinene and terpinolene in both ionization modes, whereas this was not the case for α-terpinene. We attributed this to the fact that, for α-terpinene, the most stable peroxy radicals formed are tertiary, which do not readily undergo this reaction pathway. Therefore, we do not believe that the non-detection of hydroxynitrates in this study is linked to the measurement method used. Furthermore, for phellandrenes, the major peroxy radicals formed are also expected to be tertiary, which further supports this interpretation. In fact, from all the terpenes we studied (γ-terpinene, α-terpinene, terpinolene, α-phellandrene, β-phellandrene), we conclude that the hydroxynitrates formation is not observed when the terpene has conjugated double bonds. This can be explained by the fact that the electronic delocalization will favor the most substituted (tertiary) radicals. All this information has been added in the manuscript L432-434: *"The hydroxynitrate was however observed for other monoterpenes having similar chemical structures, such as γ-terpinene and terpinolene (Fouqueau et al., 2022, 2020), whereas this was not the case for α-terpinene (Fouqueau et al., 2020). When comparing the studied terpenes (γ-terpinene, α-tepinene, terpinolene, α-phellandrene and β-phellandrene), it becomes evident that the hydroxynitrate formation is not observed in terpenes with conjugated double bonds. This can be explained by the fact that electronic delocalization favors the formation of the most substituted (tertiary) peroxy radicals, which do not readily undergo this reaction."*

In addition, the reviewer indicates that our study is not in agreement with the one from Claflin and Ziemann, as hydroxynitrates were "shown" to be formed. First, this study was performed for β-pinene and as discussed earlier, chemistry can differ. But more importantly, **no gas-phase products (including hydroxynitrates) have been detected in the study of Claflin et al.,** and **only assumptions** were made for their formation, based on the detection of products in the aerosol phase. In our opinion, our method, **based on double experimental identification of the products with PTR-MS** (in both $NO^+$ and $H_3O^+$ modes) is more reliable than the one proposed by Clafin and Ziemann **which proposes a mechanism for gas-phase chemistry without direct detection of gas-phase products. It is therefore "surprising" to see that the reviewer trusts more the results from this study which proposes a mechanism for gas-phase chemistry without direct detection of gas-phase products, rather than ours which is based on a double identification of the products with PTR-MS** (in both $NO^+$ and $H_3O^+$ modes).

6. **Figure 5. Throughout the mechanism, the authors have the alkoxy radicals reacting with O2 to form carbonyls. Justification should be given that this reaction could compete with decomposition under these conditions. The reaction of alkoxy radicals with O2 is typically negligible compared to decomposition or isomerization at room temperature (see Ziemann and Atkinson, 2012 and the references therein, also Vereecken and Peeters, 2009 and 2010). (besoin d'aide pour repondre à cette question)**

Maybe there is a misunderstanding of the mechanism proposed in Figure 5 and we should clarify it. The mechanism is not the full mechanism, and we don't claim that other reaction pathways than those indicated in Figure 5, do not occur. Also, we do not provide any branching ratio between the various alkoxy reaction pathways. This mechanism proposes explanation for the formation of the detected products. So, it cannot be deduced that isomerization of alkoxy radicals

is negligible, but only that products from isomerization were not detected. The title of the Figure 5 has been changed accordingly.

We agree that reactions of alkoxy radicals with $O_2$ are usually considered to be slower than decomposition and isomerization pathways. But these are general rules based on a very limited number of experimental data (mainly for OH chemistry), and we cannot exclude the reaction with $O_2$ from the mechanism. For other chemical systems, reaction with $O_2$ was shown to be significant (e.g. Picquet-Varrault et al., 2000). As indicated in the paper from Ziemann et Atkinson, 2012 suggested by the reviewer, *"the data-base from absolute rate methods concerning the reactions of alkoxy radicals is sparse, and at this time the relative importance of the various alkoxy radical reactions can be obtained from the use of estimation methods."* The lack of experimental data remains in 2025. So, we consider that we should not exclude any of the possible reaction pathways of the alkoxy radicals.

**7. Line 465: The definitive assignment here of carbonyl, hydroxyl, or hydroperoxide compounds is not appropriate from the methods used. For example, the molecular formula of a "carbonyl" could also correspond to an epoxide (or others) as proposed by the authors previously in the paper.**

We fully agree with the reviewer that the method used in this study (TOF-CIMS) does not allow for distinguishing between isomers with the same molecular formula, such as carbonyls and epoxides. The assignment of compound classes (e.g., $C_{10}H_{15}NO_x$ and $C_{10}H_{17}NO_x$) in our manuscript was not intended to be definitive but rather a hypothesis based on a combination of our observations and previous work in the literature on HOM formation (e.g. Dam et al., 2022; Shen et al., 2021; Guo et al., 2022).

To avoid definitive assumptions, we have revised the manuscript to clarify that, for example, we added that $C_{10}H_{15}NO_x$ could correspond to a carbonyl or epoxide (e.g. L492 and Table2- column names). As noted by Bianchi et al. (2019), "The elemental composition of epoxide products is identical to that of ketone products formed from OH loss of COOH groups, making it difficult to assess the relative importance of the two reaction routes solely based on mass."

We have replaced definitive terms with conditional language, such as "which may correspond-L492", throughout the manuscript to reflect this uncertainty.

We thank the reviewer for highlighting this important point, and we have incorporated the suggested adjustments to ensure greater accuracy and clarity in the manuscript. The text has been changed in consequence: *"The radical chain termination of $C_{10}H_{16}NO_{2n+1}^{\bullet}$ and $C_{10}H_{16}NO_{2n}^{\bullet}$ can occur through unimolecular termination channels, such as OH loss following H-abstraction from a carbon with a –OOH group attached, or through bimolecular reactions such as $RO_2 + RO_2$. These termination reactions lead to the formation of closed-shell products such as carbonyl-nitrates, hydroxynitrates and hydroperoxynitrates. It is important to note that compounds with the same molecular formula, such as hydroxynitrates formed from $C_{10}H_{16}NO_x^{\bullet}$ and hydroperoxynitrates formed from $C_{10}H_{16}NO_{x-1}^{\bullet}$, cannot be distinguished based on mass spectra analysis using TOF-CIMS. Among the $C_{10}$-HOM monomers, $C_{10}H_{15}NO_8$ and $C_{10}H_{17}NO_8$ were identified as having the highest signal intensities. Notably, the signal of $C_{10}H_{15}NO_8$ was found to be higher than that of $C_{10}H_{17}NO_8$. This has also been observed in the chemical systems involving β-pinene + $NO_3$ (Dam et al., 2022; Shen et al., 2021) and limonene + $NO_3$ (Guo et al., 2022). Finally, we also identified a family of $C_9$-HOMs ($C_9H_{14}N_2O_{9-10}$), which were detected with more intense signals for β-phellandrene than for α-phellandrene. These products can be formed by the cleavage of an exocyclic or endocyclic C-C bond followed by autoxidation. For β-phellandrene, a first-generation $C_9H_{14}N_2O_9$ compound and a second-generation $C_9H_{14}N_2O_{10}$ compound were detected but we are not able to propose detailed formation mechanisms at this stage."*

**8. Lines 488 - 492: Without measured sensitivities for these ions (which also likely consist of multiple compounds, with varying sensitivities), the detection of one molecular formula being "higher" than the other should not be used to conclude that one type of product is formed more than another, or make conclusions about specific reaction channels. Also, definitively stating that these are carbonyl / hydroxy / hydroperoxy nitrates without supplemental measurement to confirm structure or the presence of these functional groups is inappropriate.**

We agree that calibration of the $NO_3^-$ ToF-CIMS is necessary to determine its sensitivity to an organic molecule. However, to date, there is no direct calibration method capable of evaluating all possible organic compounds. Due to the lack of HOM standards, calibration is most often performed using sulfuric acid or, in some cases, other molecules, with the assumption that HOMs and the compounds used for calibration cluster with the reagent ion at similar, collision-limited rates.

Comparisons based on relative abundances are a practice already established in the literature, as demonstrated by studies such as Dam et al. (2022), Shen et al. (2021), and Guo et al. (2022). But, we fully agree with the reviewer that as no calibration was performed for each individual products, we cannot conclude on their quantification. To adopt a more cautious approach, we replaced "most abundant" with "detected with the highest intensities" in the manuscript. We made the assumption that if the compounds have similar sensitivity, comparisons based on their intensities can be considered valid.

**9. Line 530 - 545: The qualitative detection of particle phase compounds containing > 3 nitrogen containing (I assume ON) groups, is very interesting. Further exploring the formation of these compounds (multi-phase or particle-phase reactions?) would be very nice.**

We share the reviewer's opinion on the importance of exploring the formation of particle-phase compounds containing more than three nitrogen (presumably organic nitrate) groups. As mentioned in lines 550-553, we attempted to identify these compounds in the gas phase using TOF-CIMS but were unable to detect them. Online detection of the particle-phase composition using advanced mass spectrometry techniques, with online particle-phase extraction and ionization adapted to these compounds, could provide valuable insights into their real-time formation and profiles, alongside the "parental" compounds, to help elucidate their formation mechanism. This is an interesting point that must be explored in future works.

**10. Lines 536 - 539: Is this a valid assumption (same sensitivity for each compound)? What is the error? The authors need to provide some justification for the use of this, can they show the same sensitivity using proxy compounds? Otherwise the quantitative nature of this discussion should be eliminated.**

We have made modifications (section 3.5) to clarify that what we are comparing is a relative intensity and not an abundance. However, for the comparison between α- and β-phellandrenes, we believe it can be kept, even if we do not have sensitivity, since we are comparing the same products under the same operational conditions of the TOF-CIMS and the experiment.

Citation: https://doi.org/10.5194/egusphere-2024-3419-RC2

**References:**

Bianchi, F., Kurtén, T., Riva, M., Mohr, C., Rissanen, M.P., Roldin, P., Berndt, T., Crounse, J.D., Wennberg, P.O., Mentel, T.F., Wildt, J., Junninen, H., Jokinen, T., Kulmala, M., Worsnop, D.R., Thornton, J.A., Donahue, N., Kjaergaard, H.G., Ehn, M., 2019. Highly Oxygenated Organic Molecules (HOM) from Gas-Phase Autoxidation Involving Peroxy Radicals: A Key Contributor to Atmospheric Aerosol. Chem. Rev. 119, 3472–3509. https://doi.org/10.1021/acs.chemrev.8b00395

Dam, M., Draper, D.C., Marsavin, A., Fry, J.L., Smith, J.N., 2022. Observations of gas-phase products from the nitrate-radical-initiated oxidation of four monoterpenes. Atmospheric Chemistry and Physics 22, 9017–9031. https://doi.org/10.5194/acp-22-9017-2022

Fayad, L., Coeur, C., Fagniez, T., Secordel, X., Houzel, N., Mouret, G., 2021. Kinetic and mechanistic study of the gas-phase reaction of ozone with γ-terpinene. Atmospheric Environment 246, 118073. https://doi.org/10.1016/j.atmosenv.2020.118073

Fouqueau, A., Cirtog, M., Cazaunau, M., Pangui, E., Doussin, J.-F., Picquet-Varrault, B., 2022. An experimental study of the reactivity of terpinolene and $\beta$-caryophyllene with the nitrate radical. Atmospheric Chemistry and Physics 22, 6411–6434. https://doi.org/10.5194/acp-22-6411-2022

Fouqueau, A., Cirtog, M., Cazaunau, M., Pangui, E., Doussin, J.-F., Picquet-Varrault, B., 2020. A comparative and experimental study of the reactivity with nitrate radical of two terpenes: $\alpha$-terpinene and $\gamma$-terpinene. Atmospheric Chemistry and Physics 20, 15167–15189. https://doi.org/10.5194/acp-20-15167-2020

Guo, Y., Shen, H., Pullinen, I., Luo, H., Kang, S., Vereecken, L., Fuchs, H., Hallquist, M., Acir, I.-H., Tillmann, R., Rohrer, F., Wildt, J., Kiendler-Scharr, A., Wahner, A., Zhao, D., Mentel, T.F., 2022. Identification of highly oxygenated organic molecules and their role in aerosol formation in the reaction of limonene with nitrate radical. Atmospheric Chemistry and Physics 22, 11323–11346. https://doi.org/10.5194/acp-22-11323-2022

Kilpinen, O., Liu, D., Adamsen, A.P.S., 2012. Real-Time Measurement of Volatile Chemicals Released by Bed Bugs during Mating Activities. PLOS ONE 7, e50981. https://doi.org/10.1371/journal.pone.0050981

Kim, S., Karl, T., Guenther, A., Tyndall, G., Orlando, J., Harley, P., Rasmussen, R., Apel, E., 2010. Emissions and ambient distributions of Biogenic Volatile Organic Compounds (BVOC) in a ponderosa pine ecosystem: interpretation of PTR-MS mass spectra. Atmospheric Chemistry and Physics 10, 1759–1771. https://doi.org/10.5194/acp-10-1759-2010

Li, H., Riva, M., Rantala, P., Heikkinen, L., Daellenbach, K., Krechmer, J.E., Flaud, P.-M., Worsnop, D., Kulmala, M., Villenave, E., Perraudin, E., Ehn, M., Bianchi, F., 2020. Terpenes and their oxidation products in the French Landes forest: insights from Vocus PTR-TOF measurements. Atmospheric Chemistry and Physics 20, 1941–1959. https://doi.org/10.5194/acp-20-1941-2020

Maji, S., Beig, G., Yadav, R., 2020. Winter VOCs and OVOCs measured with PTR-MS at an urban site of India: Role of emissions, meteorology and photochemical sources. Environmental Pollution 258, 113651. https://doi.org/10.1016/j.envpol.2019.113651

Shen, H., Zhao, D., Pullinen, I., Kang, S., Vereecken, L., Fuchs, H., Acir, I.-H., Tillmann, R., Rohrer, F., Wildt, J., Kiendler-Scharr, A., Wahner, A., Mentel, T.F., 2021. Highly Oxygenated Organic Nitrates Formed from NO3 Radical-Initiated Oxidation of β-Pinene. Environ. Sci. Technol. 55, 15658–15671. https://doi.org/10.1021/acs.est.1c03978